# MMMR: Benchmarking Massive Multi-Modal Reasoning Tasks

## Abstract

Recent advances in Multi-Modal Large Language Models (MLLMs) have enabled unified processing of language, vision, and structured inputs, opening the door to complex tasks such as logical deduction, spatial reasoning, and scientific analysis. Despite their promise, the reasoning capabilities of MLLMs—particularly those augmented with intermediate thinking traces (MLLMs-T)—remain poorly understood and lack standardized evaluation benchmarks. Existing work focuses primarily on perception or final answer correctness, offering limited insight into how models reason or fail across modalities. To address this gap, we introduce the MMMR, a new benchmark designed to rigorously evaluate multi-modal reasoning with explicit thinking. The MMMR comprises 1) a high-difficulty dataset of 1,083 questions spanning six diverse reasoning types with symbolic depth and multi-hop demands and 2) a modular Reasoning Trace Evaluation Pipeline (RTEP) for assessing reasoning quality beyond accuracy through metrics like relevance, consistency, and structured error annotations. Empirical results show that MLLMs-T overall outperform non-thinking counterparts, but even top models like Claude-3.7-Sonnet and Gemini-2.5 Pro suffer from reasoning pathologies such as inconsistency and overthinking. This benchmark reveals persistent gaps between accuracy and reasoning quality and provides an actionable evaluation pipeline for future model development. Overall, the MMMR offers a scalable foundation for evaluating, comparing, and improving the next generation of multi-modal reasoning systems.

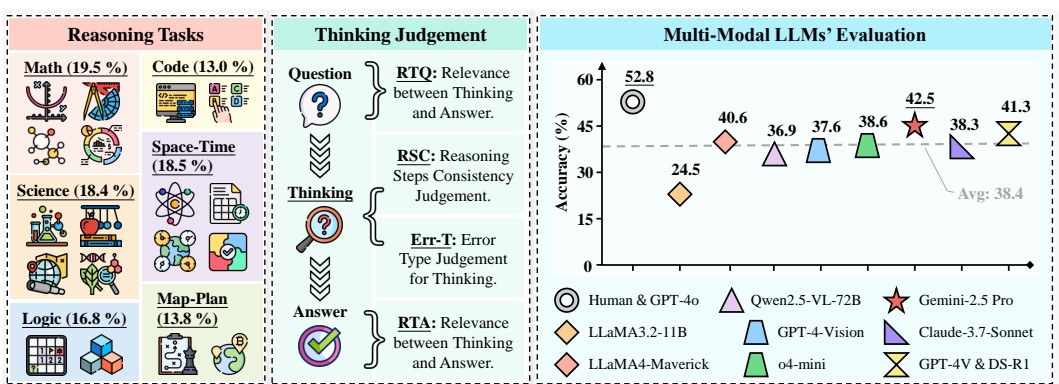

Figure 1: Overview of the Benchmark MMMR. Left: Six reasoning task types covered in the benchmark. Middle: Thinking Judgements assessing alignment for relevance (RTQ, RTA), consistency (RSC), and failure types (Err-T). Right: Accuracy distribution across models shows a clear performance gap between human experts and state-of-the-art MLLMs-T.

## 1 Introduction

The rapid advancement of Multi-Modal Large Language Models (MLLMs) has significantly enhanced unified reasoning capabilities across language, vision, and structured data modalities. Early MLLMs such as Qwen-VL Bai et al. (2023), LLaVA Liu et al. (2023a), and GPT-4 Vision OpenAI (2023) have demonstrated impressive performance in perception-centric tasks, including visual question

answering Antol et al. (2015), image captioning Sharma et al. (2018), and grounded retrieval Radford et al. (2021). However, their proficiency remains limited in tasks necessitating structured reasoning, symbolic abstraction, or sequential multi-step inference. To address this limitation, a new paradigm—MLLMs incorporating explicit intermediate reasoning (MLLMs-T)—has emerged. Representative models such as Gemini-2.5 Pro DeepMind (2025a) and Claude-3.7-Sonnet Anthropic (2025) leverage Chain-of-Thought-style reasoning, decomposing complex problems into interpretable intermediate steps, thereby emulating human-like structured problem-solving in domains such as logical deduction, scientific analysis, and code reasoning.

Despite significant progress, rigorously evaluating the reasoning capabilities of MLLMs-T remains challenging. Current benchmarks, including MMBench Liu et al. (2023e), MME-CoT Jiang et al. (2025), and MMMU Yue et al. (2023), predominantly emphasize broad coverage of tasks and perceptual understanding, offering limited insights into the reasoning process itself. Consequently, a critical research question arises: ***To what extent do MLLMs-T reliably generate coherent, interpretable, and cognitively aligned reasoning traces in complex multi-modal tasks?***

Addressing this research question demands a benchmark emphasizing reasoning depth rather than breadth. Thus, we introduce MMMR, a comprehensive benchmark explicitly designed to evaluate the multi-modal reasoning capabilities of both MLLMs and MLLMs-T. As shown in Figure 1, our benchmark comprises 1,083 rigorously curated high-difficulty tasks spanning six reasoning domains: logical reasoning Jiang et al. (2025); Xiao et al. (2024), mathematical problem-solving Lu et al. (2024); Hao et al. (2025), spatio-temporal understanding Jiang et al. (2025); Xu et al. (2024), code reasoning Li et al. (2024b;a), map-based planning Liu et al. (2024b;a), and scientific analysis Jiang et al. (2025); Yue et al. (2023).

Unlike prior benchmarks Liu et al. (2023e); Yue et al. (2023); Lu et al. (2024), MMMR introduces a structured *Reasoning Trace Evaluation Pipeline (RTEP)*, capturing reasoning trace relevance, logical consistency, and frequent error types. This structured evaluation identifies key reasoning pitfalls such as overthinking, trace inconsistency, and logical errors. The right panel of Figure 1 reveals a pronounced performance gap between state-of-the-art MLLMs-T and human-level expert reasoning. Specifically, while the best-performing model, Gemini-2.5 Pro, achieves a test accuracy of 42.45%, human experts assisted by GPT-4o reach 52.85%. This 10.3% margin underscores the persistent challenge in closing the reasoning gap, even with advanced architectures and explicit thinking mechanisms. In summary, our contributions include:

- **A comprehensive benchmark for multi-modal reasoning.** We introduce MMMR, the first benchmark that systematically targets *multi-modal reasoning* across six ing domains—Logic, Math, Code, Map, Science, and Space-Time. Unlike prior datasets, MMMR emphasizes hard question solutions and cross-modal alignment to increase reasoning complexity.

- **The first evaluation pipeline for thinking of MLLMs-T.** We propose the Reasoning Trace Evaluation Pipeline (RTEP), the first framework to incorporate intermediate *thinking trace* analysis into multi-modal reasoning evaluation. RTEP assesses reasoning relevance, stepwise consistency, and alignment, which enables deeper diagnostic insight beyond accuracy.

- **Insights into reasoning capabilities and failures.** Through extensive evaluation on MMMR, we find that state-of-the-art MLLMs-T achieve high answer accuracy across tasks, yet frequently produce flawed reasoning traces—exhibiting logical inconsistency or overthinking. These findings expose a critical misalignment between surface-level correctness and reasoning fidelity, offering new evaluation directions for future multi-modal model architecture.

## 2 MMMR: BENCHMARKING MASSIVE MULTI-MODAL REASONING TASKS

The MMMR is a challenging benchmark dataset meticulously crafted to evaluate the reasoning capabilities of Multi-modal Large Language Models with intermediate Thinking (MLLMs-T). Motivated by recent advancements in MLLMs-T, exemplified by Gemini-2.5 Pro DeepMind (2025b), which leverage intermediate reasoning processes to enhance performance, we propose a rigorous three-stage evaluation pipeline. This pipeline is specifically designed to evaluate multi-modal reasoning quality and, crucially, assess the effectiveness and robustness of intermediate thinking. As illustrated in Figure 2, our evaluation pipeline comprises three core stages.

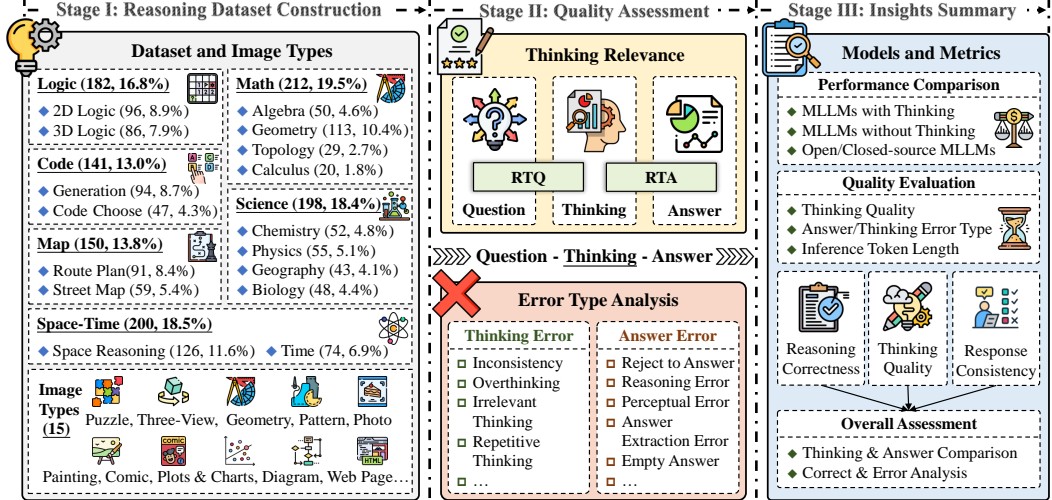

Figure 2: Overview of the MMMR evaluation pipeline. Stage I involves the creation of a challenging multi-modal reasoning benchmark dataset. Stage II evaluates the quality and structural integrity of intermediate reasoning generated by MLLMs-T. Stage III synthesizes insights regarding reasoning strategies, effectiveness, and common failure patterns across different tasks and models.

## 2.1 STAGE I: REASONING DATASET CONSTRUCTION

In this initial stage, we construct the MMMR dataset to thoroughly evaluate MLLMs-T across a wide spectrum of reasoning scenarios. The MMMR comprises 1,083 carefully curated multi-modal tasks, systematically categorized into six distinct reasoning types: Logic (16.8%), Math (19.5%), Space-Time (18.5%), Code (13.0%), Map (13.8%), and Science (18.3%). Each reasoning type further includes task-specific subcategories, such as deductive inference, algebraic calculation, temporal ordering, code generation, spatial planning, and hypothesis evaluation. The MMMR incorporates heterogeneous modalities including natural language texts, visual imagery, and structured data (e.g., Three-View diagrams, Plots & Charts, and Web Pages). To facilitate reproducible and granular evaluation, the dataset is partitioned into a validation set (106 samples) and a test set (977 samples).

## 2.2 STAGE II: THINKING QUALITY ASSESSMENT

This stage systematically evaluates the quality and structure of intermediate reasoning processes (i.e., *thinking*) produced by MLLMs-T. We propose a reasoning trace evaluation pipeline (RTEP), including metrics: 1) RTQ, quantifying the relevance of *thinking* with the posed question, and 2) RTA, assessing logical relevance between *thinking* and the answer. Both metrics are normalized within the [0,1] interval and evaluated through standardized prompts designed for precise, unbiased assessment. Furthermore, we conduct an extensive error type analysis, categorizing reasoning failures into distinct types, including thinking errors and answer errors. This fine-grained analysis offers critical insights into the strengths and vulnerabilities inherent in the reasoning approaches adopted by MLLMs-T.

## 2.3 STAGE III: REASONING INSIGHTS SYNTHESIS

The final stage synthesizes the observations from Stage II to generate holistic reasoning insights. Due to API restrictions, statistical significance tests were limited for closed-source models, consistent with Yue et al. (2023). Open-source models showed significant differences. We pursue three primary analytical objectives: 1) comparing and benchmarking the performance of MLLMs-T against standard MLLMs across different reasoning tasks, 2) profiling the quality of intermediate reasoning to identify consistent patterns of strength and weakness, and 3) investigating how prevalent error types (especially overthinking and redundant reasoning stages) impact the overall reliability of reasoning outcomes. MLLM-T evaluations used 8x NVIDIA A100 GPUs (80GB), with an average runtime of 2.5 hours for 977 test tasks. By aggregating and analyzing detailed results across various tasks and model variants, this stage supports informed interpretation of reasoning behavior.

## 2.4 RESEARCH QUESTIONS

To systematically guide our evaluation and produce insightful analyses on the reasoning capabilities of MLLMs-T, we articulate the following research questions:

> **[RQ1]** How do MLLMs-T perform in comparison to standard MLLMs concerning reasoning accuracy across the diverse and challenging multi-modal tasks presented in MMMR?
>
> **[RQ2]** How does the quality of intermediate thinking generated by MLLMs-T vary across different levels of task complexity and modality combinations?
>
> **[RQ3]** Which reasoning error types are most frequently encountered by MLLMs-T within different task contexts of MMMR, and how do these errors reflect underlying challenges in multi-modal integration?

## 3 EXPERIMENT SETTINGS

**Multi-Modal Language Models without Thinking (MLLMs).** MLLMs solve multi-modal tasks by directly mapping perception inputs to answers, bypassing explicit reasoning steps. We evaluate representative models from both open-source and closed-source. Open-source MLLMs include LLaVA-3.2-11B-Vision-Instruct Liu et al. (2023b), LLaVA-3.2-90B-Vision-Instruct Liu et al. (2023b), Qwen2.5-VL-32B-Instruct Academy (2025a), Qwen2.5-VL-72B-Instruct Academy (2025b), and Qwen-VL-max Academy (2024). Closed-source MLLMs include Gemini-1.5 Flash DeepMind (2024a), GPT-4 Vision OpenAI (2023), and LLaMA-4-Maverick AI (2025a), recognized for their sophisticated multi-modal fusion and retrieval-based response capabilities.

**Multi-Modal Language Models with Thinking (MLLMs-T).** MLLMs-T extend the capabilities of MLLMs by producing intermediate reasoning traces (Thinking) before final answer generation. We include open models such as QVQ-72B-Preview AI (2025b), and several advanced proprietary models, including Gemini-2.0 Flash Deep-Mind (2024b), Gemini-2.5 Pro DeepMind (2025a), Claude-3.7-sonnet Anthropic (2025), and o4-mini OpenAI (2025). A

Table 1: Comparison of representative multi-modal reasoning datasets. **V** (Visual Input), **OC** (Optical Characters), **I+T** (Image + Text), **IL** (Interleaved Format), **TJ** (Thinking Judgment) and **Source** (W: Web, T: Textbook, R: Remake).

| Dataset | Size | Images | Format | Source | Reason | TJ |
|---|---|---|---|---|---|---|
| VQA Antol et al. (2015) | >1M | V | I+T | W | Low | ⊗ |
| GQA Hudson and Manning (2019) | >1M | V | I+T | R | Medium | ⊗ |
| VizWiz Bigham et al. (2010) | 32K | V | I+T | W | Low | ⊗ |
| TextVQA Singh et al. (2019) | 45K | OC | I+T | W | Medium | ⊗ |
| OK-VQA Marino et al. (2019) | 14K | V+OC | I+T | W | Medium | ⊗ |
| SEED Li et al. (2024c) | 19K | V+OC | I+T | W | Medium | ⊗ |
| MMBench Liu et al. (2023e) | 3K | V+OC | I+T | W+R | Medium | ⊗ |
| MM-Vet Yu et al. (2023) | 0.2K | V+OC | I+T | W | Medium | ⊗ |
| ScienceQA Lu et al. (2022) | 6K | 5 Types | I+T | T | Medium | ⊗ |
| MME-COT Jiang et al. (2025) | 1.1K | - | IL | W+R | Medium | ⊗ |
| EMMA Hao et al. (2025) | 2.7K | - | IL | W+R | Medium | ⊗ |
| MMMU Yue et al. (2023) | 11.5K | 30 Types | IL | W+T | Medium | ⊗ |
| **MMMR** | **1.1K** | **15 Types** | **IL** | **W+T+R** | **High** | ✔ |

particularly notable configuration is our custom *Dual Model*, which simulates MLLMs-T behavior by integrating the strengths of two distinct models. Specifically, GPT-4V OpenAI (2023) is responsible for parsing the input question and image content—handling rich visual understanding and language grounding. The parsed task is then passed to DeepSeek-R1 Guo et al. (2025) for structured multi-step reasoning. This pipeline allows us to isolate and examine the effect of high-quality Thinking traces independently of perception noise. Moreover, since DeepSeek-R1 is optimized for textual reasoning rather than vision, this dual formulation ensures modular design and enhanced control over each reasoning stage.

**Datasets.** The MMMR is designed from first principles to meet the demands of benchmarking multi-modal reasoning with Thinking.Compared to prior works like MMMU Yue et al. (2023), MME-CoT Jiang et al. (2025), and EMMA Hao et al. (2025), MMMR provides a full-spectrum redesign of reasoning task settings. Each of its 1,083 problems is carefully constructed and categorized into six reasoning types and sixteen fine-grained

Table 2: Statistics of **MMMR**, detailing the distribution and characteristics of reasoning tasks.

| Category | Value |
|---|---|
| **Quantitative statistics** | |
| Total Questions | 1083 |
| Total Subjects/Subfields | 6/16 |
| Image Types | 15 |
| **Dataset Split** | |
| Validation:Test | 106:977 |
| Difficulties (Easy:Medium:Hard) | 30%:40%:30% |
| **Task Type Distribution** | |
| Logical Reasoning | 182 (16.8%) |
| Mathematical Reasoning | 212 (19.5%) |
| Spatio-Temporal Understanding | 200 (18.5%) |

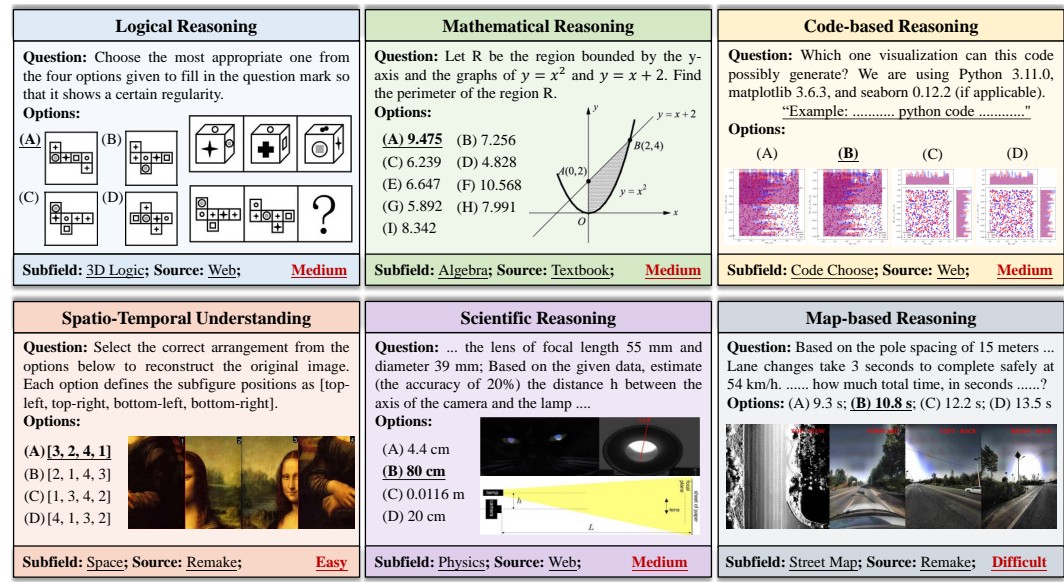

Figure 3: Representative multi-modal reasoning samples from MMMR. Each example consists of interleaved image-text input, a complex reasoning question, domain annotation, and source information. The samples illustrate the dataset's high structural variability and reasoning depth.

subfields, providing targeted coverage of logical, mathematical, spatio-temporal, code, map-based, and scientific reasoning. The dataset incorporates a rich array of visual stimuli, including charts, diagrams, 3D maps, and visual code logic, covering 15 unique image types. Unlike retrieval-centric tasks, many questions require long-horizon reasoning, abstraction, or visual-spatial synthesis. To further increase complexity, 44.6% of the items are remade or enhanced beyond web or textbook sources. Moreover, MMMR facilitates intermediate reasoning evaluation. For each sample, the source origin (Web, Textbook, Remake), and task type are documented, supporting both output-based and process-based analysis. This structure enables deep introspection into where and how reasoning fails or succeeds, making it a unique testbed for evaluating MLLMs-T.

**Metrics.** We evaluate MLLMs-T and MLLMs on the MMMR using a concise suite of metrics, with scores normalized to [0, 1]: 1) ACC, the proportion of correct answers; 2) RTQ, assessing how well the Thinking process aligns with the problem's requirements, independent of answer correctness; 3) RTA, evaluating the logical consistency between the thinking process and the final answer, regardless of accuracy; 4) Reasoning Step Consistency (RSC), measuring logical coherence across Thinking steps through consistency checks.

## 4 EMPIRICAL RESULTS AND ANALYSIS

### 4.1 MAIN RESULTS

We evaluate 17 models on the MMMR, where baselines include *Random Choice* and *Frequent Choice*, which serve as naïve heuristics, and two *Expert* configurations that represent human upper bounds with or without model assistance (see Appendix A for full descriptions). As shown in Table 3, MLLMs-T overall outperform MLLMs across six tasks, highlighting the advantage of incorporating

Table 3: Accuracy (%) comparison of baselines, MLLMs, and MLLMs-T on the MMMR benchmark. Each row highlights the per-model highest and lowest scores using green and red, respectively. For each column (task type), the best-performing model is indicated in **bold** and the second-best is underline. Models marked with * are closed-source. "S-T" denotes the Space-Time.

| | Validation (106) | Test (977) | Logic (182) | Math (212) | S-T (200) | Code (141) | Map (150) | Science (198) |
|---|---|---|---|---|---|---|---|---|
| **Baselines** | | | | | | | | |
| Random Choice | 22.1 | 23.62 | 24.18 | 24.06 | 21.50 | 25.53 | 22.67 | 23.74 |
| Frequent Choice | 26.8 | 26.58 | 26.92 | 26.42 | 24.00 | 24.82 | 25.33 | 29.80 |
| Expert (Human only) | 29.23 | - | - | - | - | - | - | - |
| Expert (Human + GPT-4o Hurst et al. (2024)) | 52.85 | - | - | - | - | - | - | - |
| **Multi-Modal Large Language Models without Thinking** | | | | | | | | |
| LLaVA-3.2-11B-Vision-Instruct Liu et al. (2023b) | 24.53 | 23.92 | 18.68 | 31.13 | 28.00 | 13.48 | 22.67 | 22.73 |
| LLaVA-3.2-90B-Vision-Instruct Liu et al. (2023b) | 30.19 | 27.65 | 21.43 | 34.91 | 35.00 | 17.73 | 25.33 | 21.72 |
| Qwen2.5-VL-32B-Instruct Academy (2025a) | 34.86 | 34.90 | 25.27 | 45.28 | 45.00 | 32.62 | 36.67 | 21.72 |
| Qwen2.5-VL-72B-Instruct Academy (2025b) | 36.95 | 37.18 | 24.18 | 46.70 | 47.50 | **41.84** | **42.67** | 31.31 |
| Qwen-VL-max Academy (2024) | 35.13 | 35.55 | 24.18 | 47.17 | 46.00 | 39.01 | 35.33 | 28.28 |
| Gemma-3-27B-IT Team et al. (2025) | 30.87 | 29.01 | 22.53 | 42.45 | 33.50 | 34.75 | 26.67 | 27.27 |
| Gemini-1.5 Flash* DeepMind (2024a) | 32.18 | 29.61 | 28.57 | 37.74 | 37.00 | 18.44 | 24.67 | 32.83 |
| GPT-4 Vision* OpenAI (2023) | 37.59 | 38.05 | 28.02 | 35.85 | 49.00 | 28.37 | 32.00 | 41.92 |
| LLaMA-4-Maverick* AI (2025a) | 40.68 | 41.82 | 30.77 | 44.81 | 46.00 | 37.59 | 30.67 | 38.38 |
| Kimi-VL-A3B-Instruct Team (2025a) | 35.2 | 34.9 | 27.6 | 35.1 | 38.4 | 32.3 | 39.5 | 40.0 |
| Llama-3.2-11B-Vision AI (2024b) | 33.5 | 33.2 | 28.9 | 32.5 | 36.8 | 31.1 | 37.9 | 34.7 |
| Qwen2.5-VL-7B Team (2025c) | 36.7 | 36.1 | 28.4 | 38.5 | 35.9 | 31.9 | 35.7 | 41.0 |
| **Multi-Modal Large Language Models with Thinking** | | | | | | | | |
| QVQ-72B-Preview AI (2025b) | 30.94 | 32.09 | 26.37 | 38.21 | 42.00 | 32.62 | 31.33 | 32.83 |
| Gemini-2.0 Flash* DeepMind (2024b) | 37.63 | 37.89 | 35.16 | **50.47** | 49.50 | 28.37 | 30.67 | 41.41 |
| Gemini-2.5 Pro* DeepMind (2025a) | **42.45** | **42.36** | **39.56** | 41.51 | 44.50 | 36.17 | 37.33 | **46.46** |
| Claude-3.7-sonnet* Anthropic (2025) | 38.28 | 37.72 | 35.71 | 45.75 | **51.00** | 21.28 | 34.00 | 43.94 |
| o4-mini* OpenAI (2025) | 38.64 | 37.58 | 34.62 | 46.23 | 47.50 | 19.86 | 29.33 | 41.41 |
| Dual (OpenAI (2023) + DeepSeek-R1 Guo et al. (2025)) | 41.26 | 41.00 | 35.71 | 47.64 | 48.00 | 22.70 | 22.67 | 45.45 |
| Kimi-VL-A3B-Thinking Team (2025b) | 35.8 | 34.5 | 28.2 | 36.4 | 39.7 | 31.9 | 40.1 | 41.3 |
| Llama-3.2V-11B-cot AI (2024a) | 34.1 | 33.8 | 29.5 | 33.7 | 37.2 | 30.4 | 38.6 | 35.9 |
| VLAA-Thinker-Qwen2.5VL-7B Team (2025d) | 37.2 | 36.4 | 29.1 | 39.8 | 35.2 | 32.6 | 36.3 | 42.1 |

explicit thinking mechanisms. Notably, *Gemini-2.5 Pro* achieves the highest overall test accuracy at 42.36%, while the *Expert (Human + GPT-4o)* attains an upper-bound of 52.85%, indicating a remaining gap between state-of-the-art MLLMs-T and human-assisted reasoning [RQ1 Summary].

**Model-wise performance reflects generalization differences.** By examining the best and second-best performers for each task column, we observe that *Gemini-2.5 Pro* (with 4 best and 1 second-best scores) exhibits the most stable and competitive accuracy across reasoning types. This suggests that explicit reasoning modules, when paired with carefully supervised thinking strategies, enable strong generalization across diverse task formats. *Gemini-2.0 Flash* performs robustly in Math (50.47%) but shows a significant drop in Code, indicating limited cross-domain transferability. The *Dual* achieves strong results in Logic and Science, supporting the effectiveness of modular architectural design. In contrast, open-source models like *Qwen2.5-VL-72B* and *Qwen-VL-max* display isolated strengths in domains such as Spatio-Temporal reasoning and Code, but lack consistency.

**Task-wise analysis reveals variation in reasoning difficulty.** A row-wise comparison of model accuracy extremes reveals task-dependent performance variability. *Math* and *Space-Time* tasks, which dominate the best-performing entries (green highlights), are generally more tractable, suggesting progress in symbolic computation and spatial comprehension. In contrast, tasks like *Logic* and especially *Code* show lower accuracy ceilings (often below 42%) and greater inter-model dispersion, as evidenced by the concentration of minimum scores (red highlights). These patterns underscore the utility of MMMR in enabling fine-grained analysis of multimodal reasoning capabilities, offering a more diagnostic lens than aggregate performance alone.

## 4.2 THINKING QUALITY ANALYSIS

To evaluate intermediate reasoning quality beyond answer correctness, we introduce the Reasoning Trace Evaluation Pipline (RTEP). This structured pipeline annotates and scores each model's reasoning trace across three dimensions: Relevance to the Question (RTQ), Relevance to the Answer (RTA), and Reasoning Step Consistency (RSC), each rated on a 0–10 scale. Scores combine rule-based checklists and semantic checks. Leveraging GPT-4o Hurst et al. (2024) as an automated evaluator,

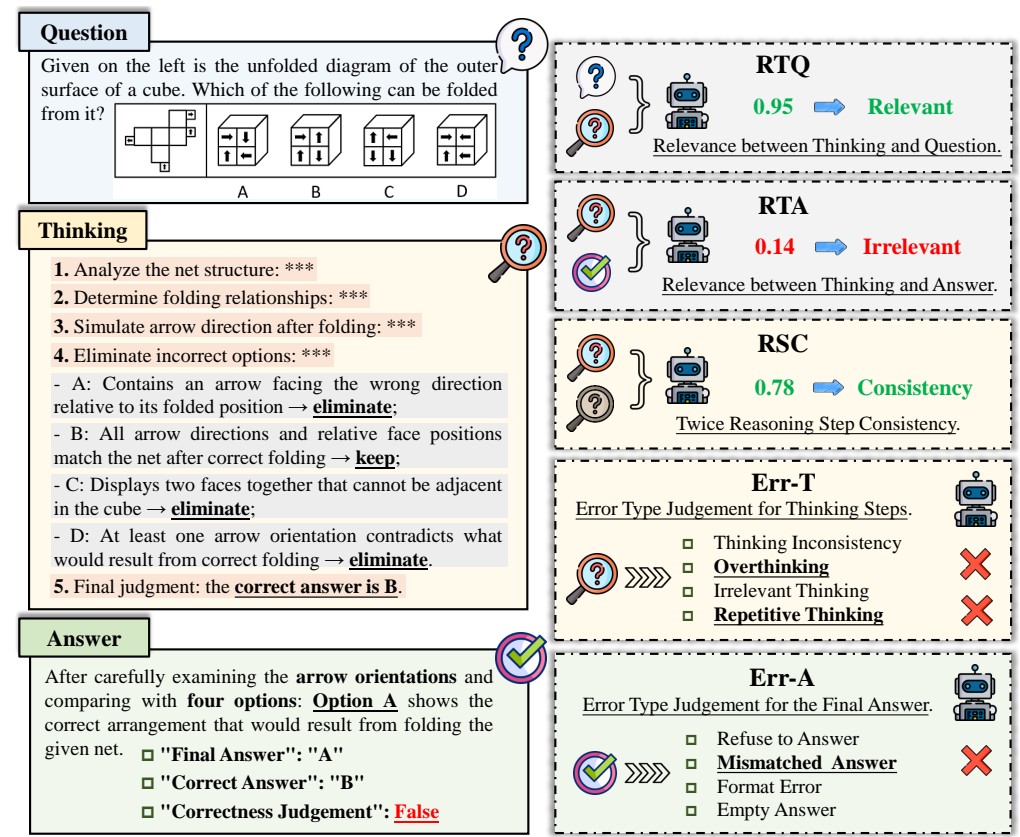

Figure 4: Overview of the Reasoning Trace Evaluation Pipeline (RTEP). The pipeline applies structured scoring of intermediate reasoning traces, evaluating consistency, relevance, and verbosity.

RTEP enables scalable, semantically aligned trace assessments, avoiding the subjectivity and cost of manual annotation. As illustrated in Figure 4, this design allows for model-agnostic diagnosis of coherence, verbosity, and reasoning alignment, enabling targeted comparisons across architectures and task types. GPT-4o's structured, rule-based evaluations, supported by Zheng et al. (2023), ensure reliability (Cohen's $\kappa = 0.80$ with human annotations. To mitigate LLM biases, we randomized task option order across all 1,083 tasks, conducted prompt sensitivity tests (maximum accuracy variance: 5.1%), and cross-validated 50 traces with Qwen-VL-Max and human annotators (88% agreement)

Table 4 summarizes a detailed comparison between Claude-3.7-sonnet and Dual (GPT-4V + DeepSeek-R1) across all six reasoning tasks in MMMR. Claude-3.7 consistently outperforms Dual in Overall Score (OS), with especially strong results in Math and Science, where compact, logically consistent traces are essential. In contrast, the Dual system achieves marginally higher answer accuracy in several tasks (e.g., Logic: +2.79%), but at the cost of reasoning coherence, as reflected in lower OS values and significantly inflated trace lengths (TLen often 3–5× higher). This indicates that longer outputs, while occasionally improving accuracy, tend to dilute reasoning relevance and introduce redundancy or inconsistency. For instance, in Code and Space-Time tasks, Dual's modular pipeline leads to repetitive or loosely linked steps, revealing that increased trace length does not ensure better reasoning quality. These findings emphasize that answer correctness alone is insufficient as a proxy for reasoning performance—models with higher accuracy can still produce flawed, verbose, or semantically incoherent rationales. Evaluating trace quality is thus essential for advancing the robustness and interpretability of MLLMs-T.

Overall, accurate answers do not guarantee sound reasoning. Our findings reveal that high-performing MLLMs-T can still produce incoherent thought processes, suggesting that future progress hinges not only on output correctness but on the quality of the reasoning path itself [RQ2 Summary].

Table 4: Comparison of reasoning quality between Claude-3.7-sonnet and Dual across six task types. RTQ, RTA, RSC are reasoning trace metrics in [0–10]; ACC is final answer accuracy (%); OS is a weighted overall score (*0.3·RTQ + 0.3·RTA + 0.3·RSC + 0.1·(ACC×0.1)*); TLen denotes trace length in thousands of tokens; ThinkErr indicates dominant reasoning flaw (defined in Section 4.3).

| Task | Model | RTQ | RTA | RSC | ACC (%) | OS | TLen (k) | ThinkErr |
|------|-------|-----|-----|-----|---------|-----|---------|----------|
| **Logic** | Claude-3.7-sonnet | 9.39 | 9.41 | 9.07 | 35.71 | 8.72 | 3.71 | Overthinking |
| | Dual | 6.32 | 7.63 | 6.18 | 38.50 | 6.42 | 15.19 | Irrelevant Thinking |
| | Δ (Dual−Claude) | - | - | - | + 2.79 | − 2.30 | + 11.48 | - |
| **Math** | Claude-3.7-sonnet | 8.88 | 9.02 | 8.40 | 45.75 | 8.35 | 5.32 | Overthinking |
| | Dual | 8.57 | 8.80 | 7.82 | 47.60 | 8.03 | 21.35 | Overthinking |
| | Δ (Dual−Claude) | - | - | - | + 1.85 | − 0.32 | + 16.03 | - |
| **Space-Time** | Claude-3.7-sonnet | 9.50 | 9.26 | 8.97 | 51.00 | 8.83 | 2.38 | Overthinking |
| | Dual | 8.50 | 8.75 | 7.60 | 48.00 | 8.32 | 14.83 | Repetitive Thinking |
| | Δ (Dual−Claude) | - | - | - | − 3.00 | − 0.51 | + 12.45 | - |
| **Code** | Claude-3.7-sonnet | 9.56 | 9.31 | 9.30 | 21.28 | 8.82 | 4.53 | Repetitive Thinking |
| | Dual | 8.61 | 8.56 | 7.94 | 22.90 | 7.93 | 17.24 | Inconsistency |
| | Δ (Dual−Claude) | - | - | - | + 1.62 | − 0.89 | + 12.71 | - |
| **Map** | Claude-3.7-sonnet | 9.17 | 8.94 | 8.43 | 23.80 | 8.57 | 1.76 | Irrelevant Thinking |
| | Dual | 7.08 | 7.42 | 6.42 | 22.50 | 6.99 | 12.43 | Repetitive Thinking |
| | Δ (Dual−Claude) | - | - | - | − 1.30 | − 1.58 | + 10.67 | - |
| **Science** | Claude-3.7-sonnet | 8.95 | 9.29 | 8.73 | 43.93 | 8.77 | 3.61 | Inconsistency |
| | Dual | 8.25 | 8.84 | 7.81 | 45.10 | 8.32 | 14.50 | Inconsistency |
| | Δ (Dual−Claude) | - | - | - | + 1.17 | − 0.45 | + 10.89 | - |

## 4.3 THINKING AND ANSWER ERROR TYPES ANALYSIS

To understand the structural weaknesses in multimodal reasoning, we analyze errors in both intermediate *Thinking* traces and final *Answer* predictions of Claude-3.7-sonnet on the MMMR validation set. As visualized in Figure 5, we classify each error into semantically distinct categories, allowing targeted diagnosis of reasoning failures.

**Thinking Errors Distribution.**

*1) Inconsistency (41.5%)* reflects internal contradictions or self-conflicting logic, often arising in Science or Logic tasks where maintaining state across steps is nontrivial.

*2) Overthinking (20.5%)* denotes unnecessarily verbose or speculative reasoning paths. These are prevalent in otherwise simple tasks where compact reasoning suffices.

*3) Irrelevant Thinking (18.5%)* includes content unrelated to the question or answer. These errors typically occur in poorly grounded inputs or under weak alignment.

*4) Repetitive Thinking (16.2%)* captures duplication without informational gain, frequently observed in Code and Map, where step-tracking or termination is difficult.

*5) Others (3.8%)* contain rare phenomena such as speculative completion or omitted critical steps.

**Answer Errors Distribution.**

*1) Reasoning Error (43.6%)* indicates logically flawed reasoning that nonetheless produces confident but incorrect answers, especially common in Math and Science.

*2) Perceptual Error (28.2%)* reflects misinterpretation of visual data such as spatial layouts or charts—frequent in Map and Space-Time tasks.

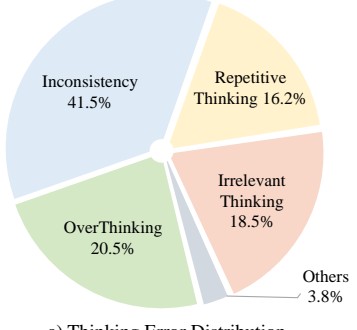

a) Thinking Error Distribution

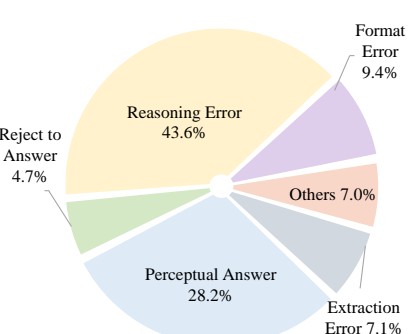

b) Answer Error Distribution

Figure 5: Distribution of Thinking and Answer Errors on Claude-3.7-sonnet.

*3) Format Error (9.4%)* denotes violations of expected
output formats, such as missing labels or extraneous text, often due to instruction-following deficiencies.

*4) Answer Extraction Error (7.1%)* occurs when models generate lengthy thinking traces but omit or fail to commit to a final answer—highlighting uncertainty or incomplete reasoning convergence.

*5) Reject to Answer (4.7%)* involves abstention despite solvable inputs, typically due to cautious decoding or alignment penalties.

*6) Others (7.0%)* include ambiguous completions or partially correct statements.

**Error Analysis.** The error distributions suggest that high answer accuracy often masks underlying reasoning path defects. The dominance of inconsistency and overthinking in reasoning traces reveals fundamental challenges in maintaining logical control and brevity. Likewise, the prevalence of reasoning-based answer errors over perceptual ones underscores that symbolic structure, rather than visual understanding, remains the primary bottleneck in high-level multimodal cognition. These findings reinforce the importance of trace-aware evaluation: coarse answer-level metrics alone cannot capture reasoning fidelity [RQ3 Summary].

## 5 RELATED WORK

**Multi-Modal Large Language Models Benchmarking.** The evaluation of multimodal reasoning has progressed from VQAv2 Antol et al. (2015), GQA Hudson and Manning (2019), and VCR Zellers et al. (2019) to broader and more specialized datasets such as TextVQA Singh et al. (2019), ScienceQA Lu et al. (2022), AI2D Kembhavi et al. (2016), and SEED Li et al. (2024c), and recent large-scale benchmarks like MMBench Liu et al. (2023e), MM-Vet Yu et al. (2023), EMMA Hao et al. (2025), MathVista Lu et al. (2024), and MMMU Yue et al. (2023). These datasets have advanced task diversity and domain specificity, yet most focus primarily on answer accuracy with limited insight into reasoning quality. While efforts such as MME-CoT Jiang et al. (2025) introduce reasoning trace annotations, they mainly rely on additional CoT designs. In contrast, our MMMR is constructed as a high-difficulty, multi-domain dataset specifically for evaluating multimodal reasoning. It not only spans six distinct reasoning types with structured task design but also supports fine-grained assessment of thinking in MLLMs-T, offering a comprehensive diagnostic standard for future multimodal models. Task difficulty was determined by human annotator scoring (Fleiss' $\kappa = 0.78$) and LLM success rates ($\geq 70\%$ easy, 30–69% medium, <30% hard), yielding a 30%:40%:30% distribution across six domains.

**Reasoning Traces and Thinking Evaluation.** In textual LLMs, reasoning trace prompting methods like Chain-of-Thought Wei et al. (2022), ReAct Yao et al. (2022), and Reflexion Shinn et al. (2023) have improved interpretability and performance through explicit step-by-step reasoning. Recent works propose evaluation tools such as RATER Liu et al. (2023d) and DECKARD Liu et al. (2023c) to assess coherence, faithfulness, and hallucination in these traces. However, such evaluations are still limited to language-only settings. Our work fills this gap by enabling trace-level reasoning evaluation within a multimodal benchmark, supporting both process- and outcome-oriented assessments.

## 6 CONCLUSION

This paper presents MMMR, a new benchmark and evaluation framework for advancing the study of multi-modal reasoning in large language models. Distinct from prior efforts that primarily emphasize perception or answer correctness, MMMR targets high-complexity, symbolic reasoning across six diverse domains, including logic, mathematics, and space-time inference. To systematically assess reasoning fidelity, we propose the Reasoning Trace Evaluation Pipeline (RTEP), which incorporates structured metrics (RTQ, RTA, RSC), length-efficiency analysis, and error-type classification to evaluate the coherence and relevance of intermediate thinking. Through extensive experiments on 17 models, we find that MLLMs-T overall outperform standard MLLMs in tasks requiring structured reasoning. Our findings suggest that improving multi-modal reasoning requires not just stronger instruction tuning or scale, but more cognitively aligned architectures that optimize for both answer correctness and thinking quality. We hope this benchmark catalyzes further research on reflective reasoning, modular cognition, and generalizable multi-modal understanding.

ETHICS STATEMENT

All authors confirm adherence to the ICLR Code of Ethics throughout this work. This study does not involve human participants in sensitive or personal contexts; instead, gold-standard annotations were obtained from professional annotators with domain expertise, who labeled paired model responses under controlled guidelines. No personally identifiable or sensitive information is included in the dataset, and all model outputs originate from publicly available LLMs or LRMs. We have taken care to report methods and results transparently, avoid misrepresentation, and disclose all relevant implementation details. Potential societal impacts were considered: automated judges may influence downstream evaluation pipelines, and our analysis highlights both benefits (scalable benchmarking, efficiency-aware routing) and risks (bias amplification, miscalibration). We emphasize that RouteJudge is intended solely for research purposes, and the framework should not be deployed in high-stakes applications without additional safeguards. The authors declare no conflicts of interest.

REPRODUCIBILITY STATEMENT

To facilitate reproducibility, we release supplementary materials accompanying this submission. These include: (i) the curated RouteJudge dataset with difficulty-aware splits across reasoning and non-reasoning domains; (ii) all evaluation code for judge benchmarking and routing strategies; and (iii) detailed instructions covering environment setup, dependency requirements, and execution steps for reproducing all experiments. Key hyperparameters (e.g., decoding settings, maximum output length, cost and latency measurement protocols) are explicitly reported in the main text, while annotation aggregation rules and error-type taxonomies are documented in the supplementary materials. Together, these resources enable independent researchers to replicate our experiments, validate reported results, and extend the framework to new models or routing policies.

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

Large Language Models (LLMs) were used to aid in the writing and polishing of the manuscript. Specifically, we used an LLM to assist in refining the language, improving readability, and ensuring clarity in various sections of the paper. The model helped with tasks such as sentence rephrasing, grammar checking, and enhancing the overall flow of the text.

It is important to note that the LLM was not involved in the ideation, research methodology, or experimental design. All research concepts, ideas, and analyses were developed and conducted by the authors. The contributions of the LLM were solely focused on improving the linguistic quality of the paper, with no involvement in the scientific content or data analysis.

The authors take full responsibility for the content of the manuscript, including any text generated or polished by the LLM. We have ensured that the LLM-generated text adheres to ethical guidelines and does not contribute to plagiarism or scientific misconduct.

# A BASIC SETTINGS

Figure 6 provides a comprehensive summary of the MMMR benchmark, including task type distributions, instance counts, and sub-category breakdowns. The benchmark consists of 1,083 multi-modal reasoning tasks spanning six high-level domains: Logic, Math, Space-Time, Code, Map, and Science. Each domain contains diverse subtypes (e.g., deductive logic, algebraic manipulation, spatial tracking, etc.), designed to probe distinct reasoning faculties. The chart further reports the proportion of each task type, ensuring balanced but realistic coverage aligned with real-world reasoning demands.

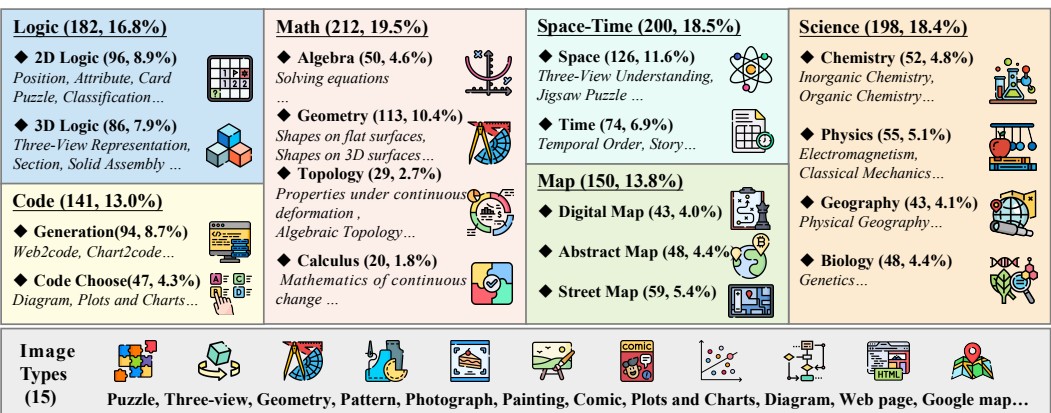

Figure 6: Task-type distribution and sub-category breakdown across the MMMR benchmark.

**Expert Baselines.** To contextualize the capabilities of MLLMs and MLLMs-T, we introduce two upper-bound baselines referred to as *Experts*, representing different degrees of human involvement:

- **Expert (Human only)**: This setting represents pure human reasoning without any model assistance. We selected three co-authors of this paper, each with graduate-level expertise in AI, cognitive science, or related fields, to independently solve the benchmark tasks. Participants were provided with full task descriptions and multi-modal inputs (text and images), but were not exposed to model outputs or allowed external tools. To ensure reliability, each sample was independently answered by at least two annotators; disagreements were resolved via majority voting. The inter-annotator agreement, measured by Krippendorff's alpha, reached 0.84, indicating high consistency and shared task understanding.

- **Expert (Human + GPT-4o)**: This hybrid configuration simulates a human-in-the-loop decision-support paradigm, where the same human experts were allowed to optionally consult GPT-4o during task resolution. Annotators first formed an independent judgment, then optionally queried GPT-4o for additional insights or solutions. Final responses reflected either acceptance or revision of GPT-4o's suggestions, along with justifications. This setting measures the upper bound of human-AI collaboration in structured reasoning tasks.

These expert configurations serve as practical performance ceilings: the human-only setting captures unaided expert cognition, while the hybrid setting reflects augmented performance with access to state-of-the-art LLM support. Together, they frame the evaluation of MLLMs within a broader continuum of human-machine reasoning capabilities.

# B PROMPT DESIGN

## B.1 BASE

**Prompt Example.** Base's prompt example is as follows:

---
**Base's prompt example**

**Zero-shot Prompt:**
►{question}
►{choice}
Please provide the final answer and store it in **\boxed{answer}**.
**Critique Prompt:** Review your previous answer and find problems with your answer.
**Improve Prompt:** Based on the problems you found, improve your answer. Please reiterate your answer, with your final answer a single numerical number, In the form **\boxed{answer}**.

---

## B.2 THINKING PROMPT

**Prompt Example.** The Thinking prompt is as follows:

---
**Thinking Prompt Example**

►{question}
►{choice}
Please think deeply before your response.
Please provide the final answer and store it in **\boxed{answer}**.

---

## B.3 IMAGE-TEXT TO TEXT PROMPT

**Prompt Example.** The Image-text to text prompt is as follows:

---
**Image-text to Text Prompt Example**

►Based on the question and the image, please summary it in pure text. Just summary the question and image as detailed as possible, no need to give the answer.

---

## B.4 RANDOM CHOICE BASELINE

**Implementation Logic.** The logic for the Random Choice baseline is as follows:

---
**Random Choice Baseline Logic**

```
def random_choice_baseline(questions, output_file):
    with open(output_file, 'w', encoding='utf-8') as f_out:
        for q in questions:
            n = len(q["choices"])
            labels = [chr(ord('A') + i) for i in range(n)]
            prediction = random.choice(labels)
            correct = normalize_answer(q["correct"])

            result = {
                "question": q.get("question", ""),
                "prediction": prediction,
                "correct": correct
            }
            f_out.write(json.dumps(result, ensure_ascii=False))
```

---

## B.5 FREQUENT CHOICE BASELINE

**Implementation Logic.** The logic for the Frequent Choice baseline is as follows:

**Frequent Choice Baseline Logic**

```python
def frequent_choice_baseline(questions, output_file):
    counter = Counter()
    for q in questions:
        correct = normalize_answer(q["correct"])
        if correct:
            counter[correct] += 1

    if not counter:
        print("No valid answers found for frequent choice baseline. Skip.")
        return

    most_common_choice, _ = counter.most_common(1)[0]

    with open(output_file, 'w', encoding='utf-8') as f_out:
        for q in questions:
            correct = normalize_answer(q["correct"])
            result = {
                "question": q.get("question", ""),
                "prediction": most_common_choice,
                "correct": correct
            }
            f_out.write(json.dumps(result, ensure_ascii=False))
```

## B.6 RANDOM CHOICE BASELINE

**Implementation Logic.** The logic for the Random Choice baseline is as follows:

**Random Choice Baseline Logic**

```python
def random_choice_baseline(questions, output_file):
    with open(output_file, 'w', encoding='utf-8') as f_out:
        for q in questions:
            n = len(q["choices"])
            labels = [chr(ord('A') + i) for i in range(n)]
            prediction = random.choice(labels)
            correct = normalize_answer(q["correct"])

            result = {
                "question": q.get("question", ""),
                "prediction": prediction,
                "correct": correct
            }
            f_out.write(json.dumps(result, ensure_ascii=False))
```

# C  CASE STUDY

**Logical-Reasoning: 2D Logic**

**Question:** *Which set does the Figure belong to? Select from A, B, and C.*
**Image:** <image 1>
**Options:**
(A)

(B)

<image 1>

(C) Neither set A nor set B

**Thinking (From Claude-3.7):**
I. Interpreting the Visual Problem
This is a pattern classification task with:
Set A (top row): 5 boxes. Set B (middle row): 5 boxes. Figure (bottom): 1 box to classify as Set A, Set B, or neither.
II. Extracting Symbolic Features
☐ Set A Characteristics:
Symbols include: stars (★), filled circles (●), empty circles (○), half-filled circles (◑), and X marks (✗). Notable variation exists in both symbol count and distribution.
☐ Set B Characteristics:
Same symbol set appears. Need to identify what differentiates Set B from Set A.
☐ Bottom Figure Composition:
2 stars (★) 2 empty circles (○) 1 half-filled circle (◑) 0 filled circles (●) 0 X symbols (✗)
III. Comparative Symbol Analysis
Counted symbol distributions per box:

Set A:                                          Set B:
A1: ★×1, ●×2, ○×3, ◑×0, ✗×2          B1: ★×0, ●×1, ○×1, ◑×1, ✗×3
A2: ★×0, ●×2, ○×3, ◑×0, ✗×1          B2: ★×2, ●×0, ○×2, ◑×1, ✗×0
A3: ★×3, ●×1, ○×2, ◑×1, ✗×1          B3: ★×2, ●×2, ○×1, ◑×0, ✗×0
A4: ★×2, ●×2, ○×2, ◑×0, ✗×0          B4: ★×1, ●×3, ○×1, ◑×2, ✗×0
A5: ★×0, ●×1, ○×3, ◑×2, ✗×0          B5: ★×1, ●×0, ○×1, ◑×0, ✗×3

The Figure:
★×2, ●×0, ○×2, ◑×1, ✗×0
Observation: The Figure's symbol composition exactly matches Set B box 2.
IV. Symbol Count Comparison
Set A: 6–8 symbols Set B: 5–7 symbols Figure: 5 symbols → fits Set B ❌
V. Conclusion
The Figure's total symbol count and exact match with Set B box 2 confirm its classification.

**Correct Answer: A**
**Response:** Looking at this problem, the goal is to determine which set the bottom figure belongs to—Set A, Set B, or neither.
The figure contains: 2 stars (★), 2 empty circles (○), 1 half-filled circle (◑), No filled circles (●) or X symbols (✗)
Set B, box 2, has an identical symbol composition, and its total symbol count (5) also matches that of the figure. In contrast, Set A boxes typically contain more symbols and different combinations.
Given the perfect match in both symbol type and count, the figure clearly **belongs to Set B.** ❌

Figure 7: Logic: Thinking Case

**Logical-Reasoning: 2D Logic**

**Question:** *Which set does the Figure belong to? Select from A, B, and C.*
**Image:** <image 1>
**Options:**
(A) [figure boxes]  <image 1>
(B) [figure boxes]
(C) Neither set A nor set B

**Correct Answer: A**
**Response:** Looking at this problem, the goal is to determine which set the bottom figure belongs to—Set A, Set B, or neither.
The figure contains: 2 stars (★), 2 empty circles (○), 1 half-filled circle (◑), No filled circles (●) or X symbols (✗)
Set B, box 2, has an identical symbol composition, and its total symbol count (5) also matches that of the figure. In contrast, Set A boxes typically contain more symbols and different combinations.
Given the perfect match in both symbol type and count, the figure clearly **belongs to Set B.**

Figure 8: 2D Logic: Non-Thinking Case

---

## Logical Reasoning: 3D Logic

**Question:** *For the following three-dimensional figures, which of the four options cannot be its view (front view, top view, side view)?*
**Image:** <image 1>
**Options:**

(A)    (B)    (C)    (D)    <image 1>

---

**Correct Answer: D**
**Response:** I'll analyze this problem step by step to determine which of the four options cannot be a valid view of the three-dimensional figure shown at the top.

Let me examine the 3D figure at the top of the image. It appears to be composed of cubic blocks arranged in a specific configuration with black and white triangular patterns on each visible face.

To solve this problem, I need to mentally visualize what this 3D object would look like from different viewing angles (front, top, and side views) and compare these with the four options labeled A, B, C, and D.

When I examine a 3D cube-based object from orthogonal directions:

Each face should show a distinct 2D projection

The patterns on visible faces should match the black and white triangular patterns shown in the original figure

The overall shape should be consistent with the 3D arrangement of blocks

Looking carefully at each option:

Option A: This shows an L-shaped arrangement with specific black/white triangular patterns that could represent a valid view of the 3D figure.

Option B: This shows a rectangular arrangement with triangular patterns that could represent a valid view from certain angles.

Option C: This shows a T-shaped arrangement with triangular patterns.

Option D: This shows a zigzag arrangement with triangular patterns.

When I visualize the 3D structure from various angles, I notice that option D shows a pattern that cannot be derived from the given 3D figure. The specific arrangement of blocks and triangular patterns in option D is inconsistent with any possible orthographic projection (front, top, or side) of the 3D structure shown.

Therefore, **option D cannot be a valid view of the three-dimensional figure.**

Figure 9: 3D Logic: Non-Thinking Case

**Code: Code Generation**

**Question:** *This is a matrix traversal problem where we need to:*
*- Start from the top-right corner of a given matrix- Follow a specific spiral pattern in counter-clockwise direction*
*- Collect all elements in the order of traversal- The traversal pattern should work for matrices of different sizes (M x N)*
*# Visual Facts:*
*1.Four example matrices are shown with different dimensions:*
*- 4x1 (top-right) - 4x2 (top-left)*
*- 4x3 (bottom-left) - 4x4 (bottom-right)*
*......*
*# Visual Patterns:*
*1. Movement Direction Pattern:......*
*2. Traversal Rules:......*
*Please provide the final answer and store it in \\boxed{answer}.*
**Image:** <Image 1>

<Image 1>

**Thinking:** This problem is asking me to write code for a spiral matrix traversal, starting from the top-right corner and moving in a counter-clockwise direction.From the visual examples and description, I can see that:
1. We start at the top-right corner (0, N-1)
2. Initial direction is leftward
3. When we reach a boundary or a visited cell, we change direction in a counter-clockwise manner
4. The direction sequence is: Left $\rightarrow$ Down $\rightarrow$ Right $\rightarrow$ Up $\rightarrow$ Left $\rightarrow$ ...
5. We need to collect all elements in the order of traversal
6. The algorithm should work for any MxN matrix
Let me design an algorithm for this:……
Here's a pseudo-code representation:……

**Correct Answer:** from typing import List ……
**Response:**
# Matrix Spiral Traversal Algorithm:Based on the problem description and visual examples, I need to create an algorithm that traverses a matrix in a counter-clockwise spiral starting from the top-right corner.
## Solution:The key is to follow the directional pattern (Left $\rightarrow$ Down $\rightarrow$ Right $\rightarrow$ Up) and change direction whenever we hit a boundary or previously visited cell.

```python
def spiral_traverse(matrix):
    …..
```
### Example:
For a 3x3 matrix:```\n1 2 3\n4 5 6\n7 8 9\n```

Figure 10: Code: Thinking Case

## Code: Code Generation

**Question:** *This is a matrix traversal problem where we need to:*
*- Start from the top-right corner of a given matrix- Follow a specific spiral pattern in counter-clockwise direction*
*- Collect all elements in the order of traversal- The traversal pattern should work for matrices of different sizes (MxN)*
*# Visual Facts:*
*1.Four example matrices are shown with different dimensions:*
*- 4x1 (top-right) - 4x2 (top-left)*
*- 4x3 (bottom-left) - 4x4 (bottom-right)*
*......*
*# Visual Patterns:*
*1. Movement Direction Pattern: ......*
*2. Traversal Rules: ......*
*Please provide the final answer and store it in \\boxed{answer}.*
**Image:** <Image 1>

<Image 1>

**Correct Answer:** from typing import List ......
**Response:**
# Matrix Spiral Traversal Algorithm:Based on the problem description and visual examples, I need to create an algorithm that traverses a matrix in a counter-clockwise spiral starting from the top-right corner.
## Solution:The key is to follow the directional pattern (Left \u2192 Down \u2192 Right \u2192 Up) and change direction whenever we hit a boundary or previously visited cell.
```python
def spiral_traverse(matrix):
.....
```
### Example:
For a 3x3 matrix:```\n1 2 3\n4 5 6\n7 8 9\n```
......

Figure 11: Generation: Non-Thinking Case

## Code: Code Choose

**Question:** *Which visualization can this code possibly generate? We are using Python version 3.11.0, matplotlib version 3.6.3, and seaborn version 0.12.2 (if applicable).*
*import matplotlib.pyplot as plt*
*......*
*provide the final answer and store it in \\boxed{answer}.*
**Options:**

**Correct Answer:D**
**Response:** Visualization Analysis:Looking at the Python code and comparing it with the provided images, I can determine which visualization this code would generate.The code creates an error bar plot. Among the provided images, the **first image** most closely matches what this code would generate. It accurately shows:- The correct line styles (dashed for AD and CV, dash-dot for FCV and IWCV)- The proper markers (circles, triangles, squares, and x marks)- Error bars with the specified cap size- The exact title, axis labels, and legend positioning.The other images have either slight differences in line styling, marker appearance, or rendering issues (like the last image which has text corruption at the bottom).

Figure 12: Code Choose: Non-Thinking Case

## Math: Calculus

**Question**: *<image 1> Find the surface area bounded by the curves $y=2^x$, $y=2^{2^x}$ and $y=4$.*

**Image**: <image 1>

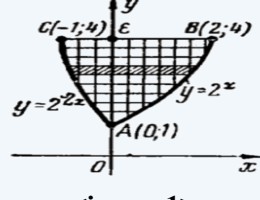

<image 1>

**Thinking (From Claude-3.7):**

I.  Identify Intersection Points
    The intersections are found to be A(0,1) from $y=2^x$ and $y=2^{-2x}$; B(2,4) from $y=2^x$ and $y=4$; and C(-1,4) from $y=2^{-2x}$ and $y=4$.

II. Define the Integration Strategy and Set Up Integrals
    The lower boundary is $y=2^{-2x}$ for $-1 \leq x \leq 0$ and $y=2^x$ for $0 \leq x \leq 2$, leading to: Area=$\int_0^{-1} 4 - 2^{-2x}\,dx + \int_0^2 4 - 2^x\,dx$.

III. Evaluate the Integrals
    The first integral solved by u-substituting $2^{-2x}$ term, resulting $4-3/(2\ln(2))$
    The second integral is computed with result of $8-3/\ln(2)$

IV. Finalize the Answer
    Summing up the two integrals gives the total area as $12 - 9/(2\ln(2))$ ❌

**Correct Answer: $24 - 9/(2\ln(2))$**

**Response:** To find the area bounded by the curves $y=2^x$, $y=2^{-2x}$, and $y=4$, the first step was to identify their intersection points, which were determined to be A(0,1), B(2,4), and C(-1,4). These points define the vertices of the region. The area calculation was then set up by dividing the region at $x=0$, recognizing that $y=4$ is the upper boundary, while the lower boundary changes from $y=2^{-2x}$ (for x between -1 and 0) to $y=2^x$ (for x between 0 and 2). This led to expressing the total area as the sum of two definite integrals: $\int_0^{-1} 4 - 2^{-2x}\,dx + \int_0^2 4 - 2^x\,dx$. The first integral, corresponding to the left portion, was evaluated to $4-3/(2\ln(2))$, involving a u-substitution for the exponential term. The second integral, for the right portion, was evaluated to $8-3/(2\ln(2))$. Finally, summing these two results yielded the total enclosed area as $12 - 9/(2\ln(2))$. ❌

Figure 13: Math: Thinking Case

**Math: Algebra**

**Question:** *Which of the following graphs are not isomorphic?*
**Options:**

(A)          (B)          (C)

**Correct Answer: A**
**Response: A, B, C**
Isomorphism requires a one-to-one mapping between vertices that preserves adjacency.
Graph 1 is described as a wheel graph with a central hub, where each of the 8 vertices has a degree of 3, making it a regular graph.
Graph 2 has a more rectangular structure with diagonals. Its vertices have degrees of either 3 or 4, resulting in a degree sequence of {3, 4, 3, 4, 4, 3, 4, 3}. This differing degree sequence immediately indicates that Graph 2 is not isomorphic to Graph 1 or Graph 3.
Graph 3 forms a square with internal vertices. All 8 of its vertices have a degree of 3, resulting in a regular graph with a degree sequence of {3, 3, 3, 3, 3, 3, 3, 3}.
Further comparison between Graph 1 and Graph 3 reveals structural differences. Graph 1's wheel-like structure with a central hub maintains connectivity upon removal of any single vertex. In contrast, removing specific vertices in Graph 3's square-based structure would disconnect the graph. This difference in connectivity properties confirms that Graph 1 and Graph 3 are not isomorphic.
Therefore, based on their degree sequences and structural properties, none of the three graphs are isomorphic to each other.
**Final Answer: A, B, C**

Figure 14: Algebra: Non-Thinking Case

## Math: Geometry

**Question:** *A $4 \times 4 \times h$ rectangular box contains a sphere of radius $2$ and eight smaller spheres of radius $1$. The smaller spheres are each tangent to three sides of the box, and the larger sphere is tangent to each of the smaller.*

**Image:** <image 1>

**Options:**

(A) $\langle a, b, c, d, e, f \mid aba^{-1}b^{-1}cdc^{-1}d^{-1}efe^{-1}f^{-1} = 1 \rangle$

(B) $\langle a, b, c, d, e, f \mid abcd = ef, a = b^{-1}, c = d^{-1}, e = f^{-1} \rangle$

(C) $\langle a, b, c, d, e, f \mid abcdef = 1 \rangle$

(D) $\langle a, b, c, d, e, f \mid ab^{-1}cd^{-1}ef^{-1} = 1 \rangle$

<image 1>

**Correct Answer: B**
**Response: (C)**

The left figure shows a surface with a specific topological structure, and the right figure appears to be the boundary of this surface or a representation of loops that define the fundamental group.

When working with fundamental groups of surfaces using presentations, we need to identify how the boundary components are glued together. The standard approach is to understand how the edges are identified in pairs.

From the diagram, I can see that we have loops labeled a through f. The arrows indicate direction, and the identifications form the surface boundary.

In the canonical presentation of the fundamental group of a surface, the product of all commutators (for orientable surfaces) or all squares (for non-orientable surfaces) equals the identity. Alternatively, if we're looking at a surface with boundary, we'd have a relation that the product of all boundary elements equals the identity. Looking at the right diagram, it appears to show a polygon with edges labeled a through f, which when glued according to the labels and orientations, forms the surface on the left. The boundary word reading clockwise is *abcdef*. For a surface with a single boundary component, as this appears to be, the fundamental group would have the relation that the product of all boundary edges (read in order) equals the identity. Therefore, the presentation that correctly represents this surface is: $\langle$ *a, b, c, d, e, f* | *abcdef* = 1$\rangle$ .

This corresponds to **option (C)**.

Figure 15: Geometry: Non-Thinking Case

## Math: Topology

**Question:** *Considering the provided diagram depicting a surface and its associated fundamental group, which of the following group presentations accurately represents the fundamental group of the surface shown in the left part of the diagram?*

**Image:** <image 1>

<image 1>

**Options:**

(A) $\langle a, b, c, d, e, f \mid aba^{-1}b^{-1}cdc^{-1}d^{-1}efe^{-1}f^{-1} = 1 \rangle$

(B) $\langle a, b, c, d, e, f \mid abcd = ef, a = b^{-1}, c = d^{-1}, e = f^{-1} \rangle$

(C) $\langle a, b, c, d, e, f \mid abcdef = 1 \rangle$

(D) $\langle a, b, c, d, e, f \mid ab^{-1}cd^{-1}ef^{-1} = 1 \rangle$

**Correct Answer: B**

**Response: (C)**

The left figure shows a surface with a specific topological structure, and the right figure appears to be the boundary of this surface or a representation of loops that define the fundamental group.

When working with fundamental groups of surfaces using presentations, we need to identify how the boundary components are glued together. The standard approach is to understand how the edges are identified in pairs.

From the diagram, I can see that we have loops labeled a through f. The arrows indicate direction, and the identifications form the surface boundary.

In the canonical presentation of the fundamental group of a surface, the product of all commutators (for orientable surfaces) or all squares (for non-orientable surfaces) equals the identity. Alternatively, if we're looking at a surface with boundary, we'd have a relation that the product of all boundary elements equals the identity. Looking at the right diagram, it appears to show a polygon with edges labeled a through f, which when glued according to the labels and orientations, forms the surface on the left. The boundary word reading clockwise is *abcdef*. For a surface with a single boundary component, as this appears to be, the fundamental group would have the relation that the product of all boundary edges (read in order) equals the identity. Therefore, the presentation that correctly represents this surface is: $\langle$ *a, b, c, d, e, f | abcdef = 1* $\rangle$ This corresponds to **option (C)**.

Figure 16: TopoLogy: Non-Thinking Case

## Math: Calculus

**Question**: *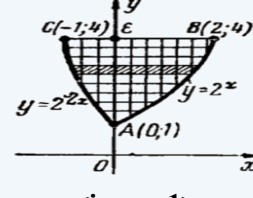 Find the surface area bounded by the curves y=2ˣ, y=2²^x and y=4.*

**Image:** <image 1>

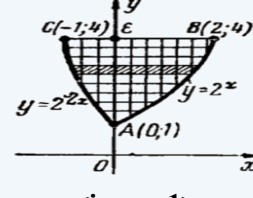

<image 1>

**Correct Answer: 24 – 9/(2ln(2))**

**Response:** To find the area bounded by the curves $y=2^x$, $y=2^{-2x}$, and y=4, the first step was to identify their intersection points, which were determined to be A(0,1), B(2,4), and C(-1,4). These points define the vertices of the region. The area calculation was then set up by dividing the region at x=0, recognizing that y=4 is the upper boundary, while the lower boundary changes from $y=2^{-2x}$ (for x between -1 and 0) to $y=2^x$ (for x between 0 and 2). This led to expressing the total area as the sum of two definite integrals: $\int_0^{-1} 4-2^{-2x}\,dx + \int_0^2 4 - 2^x\,dx$. The first integral, corresponding to the left portion, was evaluated to 4-3/(2ln(2)), involving a u-substitution for the exponential term. The second integral, for the right portion, was evaluated to 8-3/(2ln(2)). Finally, summing these two results yielded the total enclosed area as 12 – 9/(2ln(2)).

Figure 17: Calculus: Non-Thinking Case

## Space-Time: Space

**Question:** *Given the front view and left view of a solid, please choose the correct top view from options A, B, C, and D.*

**Image:** <image 1>

**Options:**

(A)     (B)     (C)     (D)

<image 1>

**Thinking (From Claude-3.7):**

I. Interpreting Provided Views

Front View (left): ***. Left View (right): ***.

II. Extracting Critical Features

Front View Observations: ***. Left View Observations: ***.

III. Reconstructing the 3D Object

Mapping Curvature from Front View: ***. Validating Depth from Left View: ***.

IV. Evaluating Top View Candidates

☐ Option A features a uniform L-shape but fails to account for the width gradation observed in the oval profile of the front view. → **eliminate**;

☐ Option B adopts a horizontally expanded L-configuration, reflecting the oval's maximum width at its horizontal centerline as depicted in the front view. → **keep**;

☐ Option C incorporates a stepped horizontal edge, introducing geometric complexity not supported by the original front or left views. → **eliminate**;

☐ Option D proposes a narrowed horizontal base, which directly conflicts with the oval's widening trend visible in the front view. → **eliminate**.

V. Concluding the Analysis

**The answer is B.**

**Correct Answer: B**

**Response:** For the correct top view, it must be consistent with both the front and left views; The vertical centerline should align with the same features in all views; The shape must account for the oval profile seen in the front view.

The oval in the front view indicates that when viewed from the top, we should see a part with varying width. Option B matches this expectation because: 1) Its wider horizontal section corresponds to the wider part of the oval in the front view. 2) The straight vertical line on the left matches the flat profile seen in the left view. 3) The overall L-shape explains how the object can appear oval from the front while rectangular from the left. **Therefore, the correct top view is B**.

Figure 18: Space-Time: Thinking Case

## Space-Time: Space

**Question:** *<image 1>Select the correct arrangement from the options below to reconstruct the original image. Each option defines the subfigure positions as [top-left, top-right, bottom-left, bottom-right].*

**Image:** <image 1>

**Options:**

A. [4, 2, 1, 3], B. [2, 3, 1, 4],
C. [1, 3, 4, 2], D. [4, 1, 3, 2].

<image 1>

**Correct Answer: D**

**Response (From Qwen-VL-Max):** To determine the correct arrangement of the subfigures, we need to analyze the content and logical placement of each subfigure in the context of a typical auditorium layout.

Subfigure 1: This image shows a close-up view of the seating area with steps leading up to the seats. It appears to be a detailed view of the lower section of the auditorium.

Subfigure 2: This image shows an elevated view of the auditorium, capturing both the seating area and the ceiling structure. It seems to be taken from a higher vantage point, possibly from the balcony.

Subfigure 3: This image focuses on the ceiling and upper part of the auditorium, showing the lighting and architectural details of the ceiling.

Subfigure 4: This image provides a broader view of the seating area, similar to Subfigure 1 but from a slightly different angle and with more of the floor visible.

Given these observations, we can deduce the following:

Top-left (position 1): This should be a view that provides a good overview but is not the most detailed or highest vantage point. Subfigure 4 seems suitable here as it gives a broad view of the seating area.

Top-right (position 2): This should be a higher vantage point view, which is best represented by Subfigure 2.

Bottom-left (position 3): This should be a detailed view of a specific part, such as the ceiling or a section of the seats. Subfigure 3, which shows the ceiling details, fits well here.

Bottom-right (position 4): This should be a detailed view of the seating area, which is best represented by Subfigure 1.

Thus, the correct arrangement is [4, 2, 3, 1].

However, the options provided are:A. [4, 2, 1, 3] B. [2, 3, 1, 4] C. [1, 3, 4, 2] D. [4, 1, 3, 2]

Comparing our deduced arrangement [4, 2, 3, 1] with the given options, the closest match is Option A: [4, 2, 1, 3].

**Therefore, the final answer is  A.**

Figure 19: Space Reasoning: Non-Thinking Case

**Space-Time: Time**

**Question:** *Which frame appears first on the match timeline?*
**Options:**

(A)    (B) 

**Correct Answer: B**
**Response (From Qwen-VL-Max):** To determine which frame appears first on the match timeline, we need to analyze the sequence of events in both images.
Image A:
The ball is in the air, approaching the goal.The goalkeeper is preparing to make a save.Players are positioned around the goal area, with some running towards the ball.
Image B:
The ball is on the ground, closer to the goalkeeper.The goalkeeper appears to be in the process of making a save or has just made contact with the ball.Players are still around the goal area, but their positions suggest a slightly earlier moment in the action compared to Image A.
In Image B, the ball is on the ground and closer to the goalkeeper, indicating an earlier moment in the sequence.In Image A, the ball is in the air, suggesting it has already been kicked and is approaching the goal, which occurs after the scenario in Image B.
**Therefore, Image B appears first on the match timeline.**

Figure 20: Time: Non-Thinking Case

## Map: Route Plan

**Question:** *How to get to Sierra Leone Grammer School from Seabird Water Taxi after having lunch at Cole Street Guesthouse?Please provide the final answer and store it in \\boxed{answer}.*

**Image:** <image 1>

<image 1>

**Options:**
**A.**Seabird Water Taxi.Head northwest on High Broad St toward Andrews ….
**B.** Seabird …St.Turn left onto Andrews St\nContinue onto Andrew St.Turn right onto Macaulay St.Sierra Leone Grammar School
**C.** …Leone Grammar School.Head northeast on Macaulay St.Turn left at the 1st cross street toward Cole St.Turn left at the 2nd cross street onto Cole St.Cole Street Guesthouse
**D.** …Cole St toward Damba Rd.Turn right at the 1st cross street toward Macaulay St.Turn right at the 2nd cross street onto Macaulay St.Sierra Leone Grammar School

**Thinking(From Claude-3.7):**
Let me analyze each option to see which describes this specific journey:
**Option 1** includes:- Directions from Seabird Water ...
**Option 2** only includes:- Directions from Seabird ….
**Option 3** includes:- Directions from Seabird ….
**Option 4** includes:- Directions from Seabird Water Taxi to Cole Street Guesthouse-Followed by directions from Cole Street Guesthouse to Sierra Leone Grammar School.
**This is the route we need!** The second part describes how to get from Cole Street Guesthouse to Sierra Leone Grammar School.

**Correct Answer:D**
**Response:**
# Directions from Cole Street Guesthouse to Sierra Leone Grammar School.Looking at the map and analyzing the provided options, I need to determine the correct route from Cole Street Guesthouse to Sierra Leone Grammar School (after having lunch at Cole Street Guesthouse).Based on the map, ...These directions provide the most efficient route from Cole Street Guesthouse to Sierra Leone Grammar School according to the map layout.

Figure 21: Map: Thinking Case

## Map: Route Plan

**Question:** *How to get to Sierra Leone Grammer School from Seabird Water Taxi after having lunch at Cole Street Guesthouse?Please provide the final answer and store it in \\boxed{answer}*.

**Image:** <image 1>

<image 1>

**Options:**
**A.**Seabird Water Taxi.Head northwest on High Broad St toward Andrews ….
**B.** Seabird …St.Turn left onto Andrews St\nContinue onto Andrew St.Turn right onto Macaulay St.Sierra Leone Grammar School
**C.** …Leone Grammar School.Head northeast on Macaulay St.Turn left at the 1st cross street toward Cole St.Turn left at the 2nd cross street onto Cole St.Cole Street Guesthouse
**D.** …Cole St toward Damba Rd.Turn right at the 1st cross street toward Macaulay St.Turn right at the 2nd cross street onto Macaulay St.Sierra Leone Grammar School

**Thinking(From Claude-3.7):**
Let me analyze each option to see which describes this specific journey:
**Option 1** includes:- Directions from Seabird Water ...
**Option 2** only includes:- Directions from Seabird ….
**Option 3** includes:- Directions from Seabird ….
**Option 4** includes:- Directions from Seabird Water Taxi to Cole Street Guesthouse-Followed by directions from Cole Street Guesthouse to Sierra Leone Grammar School.
**This is the route we need!** The second part describes how to get from Cole Street Guesthouse to Sierra Leone Grammar School.

**Correct Answer:D**
**Response:**
# Directions from Cole Street Guesthouse to Sierra Leone Grammar School.Looking at the map and analyzing the provided options, I need to determine the correct route from Cole Street Guesthouse to Sierra Leone Grammar School (after having lunch at Cole Street Guesthouse).Based on the map, ...These directions provide the most efficient route from Cole Street Guesthouse to Sierra Leone Grammar School according to the map layout.

Figure 22: Route Plan: Non-Thinking Case

## Map: Route Plan

**Question:** *How to get to Sierra Leone Grammer School from Seabird Water Taxi after having lunch at Cole Street Guesthouse?Please provide the final answer and store it in \\boxed{answer}.*

**Image:** <image 1>

<image 1>

**Options:**
A.Seabird Water Taxi.Head northwest on High Broad St toward Andrews ….
B. Seabird …St.Turn left onto Andrews St\nContinue onto Andrew St.Turn right onto Macaulay St.Sierra Leone Grammar School
C. …Leone Grammar School.Head northeast on Macaulay St.Turn left at the 1st cross street toward Cole St.Turn left at the 2nd cross street onto Cole St.Cole Street Guesthouse
D. …Cole St toward Damba Rd.Turn right at the 1st cross street toward Macaulay St.Turn right at the 2nd cross street onto Macaulay St.Sierra Leone Grammar School

**Thinking(From Claude-3.7):**
Let me analyze each option to see which describes this specific journey:
**Option 1** includes:- Directions from Seabird Water ...
**Option 2** only includes:- Directions from Seabird ….
**Option 3** includes:- Directions from Seabird ….
**Option 4** includes:- Directions from Seabird Water Taxi to Cole Street Guesthouse-Followed by directions from Cole Street Guesthouse to Sierra Leone Grammar School.
**This is the route we need!** The second part describes how to get from Cole Street Guesthouse to Sierra Leone Grammar School.

**Correct Answer:D**
**Response:**
# Directions from Cole Street Guesthouse to Sierra Leone Grammar School.Looking at the map and analyzing the provided options, I need to determine the correct route from Cole Street Guesthouse to Sierra Leone Grammar School (after having lunch at Cole Street Guesthouse).Based on the map, ...These directions provide the most efficient route from Cole Street Guesthouse to Sierra Leone Grammar School according to the map layout.

Figure 23: Street Map: Non-Thinking Case

## Science: Physics

**Question:** *A block ◆ is connected to two unstretched springs with spring constants ◆ and 4◆ . The block is displaced a distance ◆ to the left from equilibrium and released. It then moves a maximum distance y to the right. Find the ratio y/x.*

**Image:** <image 1>

**Options:**

(A) 4     (B) 2     (C) 1/2     (D) 1/4

<image 1>

**Thinking (From Claude-3.7):**

I. Interpreting Provided Views

Figure I shows the system at equilibrium with both springs unstretched. Figure II shows block B displaced a distance x toward wall 1.

II. Extracting Critical Features

Supports M1 and M2 are massless and free to move, so they adjust to keep the net force on them zero. The energy is entirely stored in the springs when the block is displaced.

III. Reconstructing the System Dynamics

At displacement x:

$$E_{initial} = (1/2) * k * x^2 + (1/2) * (4k) * x^2 = (5/2) * k * x^2$$

At maximum displacement y on the other side:

$$E_{final} = (5/2) * k * y^2$$

Conservation of energy gives:

$$(5/2) * k * x^2 = (5/2) * k * y^2 \rightarrow y = x$$

IV. Evaluating Physical Consistency

The system is isolated with no external forces; the center of mass remains stationary. Since the supports are massless, their movement does not contribute to the system's inertia or momentum. Thus, the motion is symmetric, and the block oscillates equally about the equilibrium point.

V. Concluding the Analysis

The answer is: y/x = 1.

**Correct Answer: 1/2**

**Response**: The system consists of a block connected to two springs: one with spring constant k and the other with 4k. The supports can move freely, and the system is frictionless. When the block is displaced by a distance x toward wall 1 and released, the total potential energy stored in both springs is: E_initial = $(1/2) \cdot k \cdot x^2 + (1/2) \cdot 4k \cdot x^2 = (5/2) \cdot k \cdot x^2$.

At the maximum displacement y on the opposite side, both springs are again stretched (or compressed), and the total energy is: E_final = $(1/2) \cdot k \cdot y^2 + (1/2) \cdot 4k \cdot y^2 = (5/2) \cdot k \cdot y^2$. By conservation of mechanical energy (since there is no friction), $(5/2) \cdot k \cdot x^2 = (5/2) \cdot k \cdot y^2 \rightarrow x^2 = y^2$ $\rightarrow y = x$. **Thus, the ratio y/x = 1.**

Figure 24: Science: Thinking Case

## Science: Chemistry

**Question:** *The transition-state structure changes after electron relocation or reaction. Which of the following options shows the structure after the change?*

Image: <image_1>

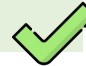

**Options:**

(A)

(B)

(C)

(D)

**Correct Answer: D**

**Response**: For the correct product after the deprotonation step in Image 1, we must identify the immediate consequence of removing the alpha hydrogen from a carbonyl compound. The carbonyl (C=O) group adjacent to an alpha hydrogen undergoes deprotonation by a base, forming an enolate ion. This ion has a delocalized negative charge between the oxygen of the carbonyl and the alpha carbon. Image 5 correctly represents this enolate structure: 1) The negative charge is shown on the oxygen, consistent with resonance stabilization. 2) The carbon-carbon double bond has shifted accordingly to reflect enolate formation. 3) Other options either show further reaction products (Image 2 and Image 3) or the unchanged starting material (Image 4).
Therefore, **the correct immediate product after deprotonation is shown in D**.

Figure 25: Chemistry: Non-Thinking Case

## Science: Physics

**Question:** *A block ◆ is connected to two unstretched springs with spring constants ◆ and 4◆ . The block is displaced a distance ◆ to the left from equilibrium and released. It then moves a maximum distance y to the right. Find the ratio y/x.*

**Image:** 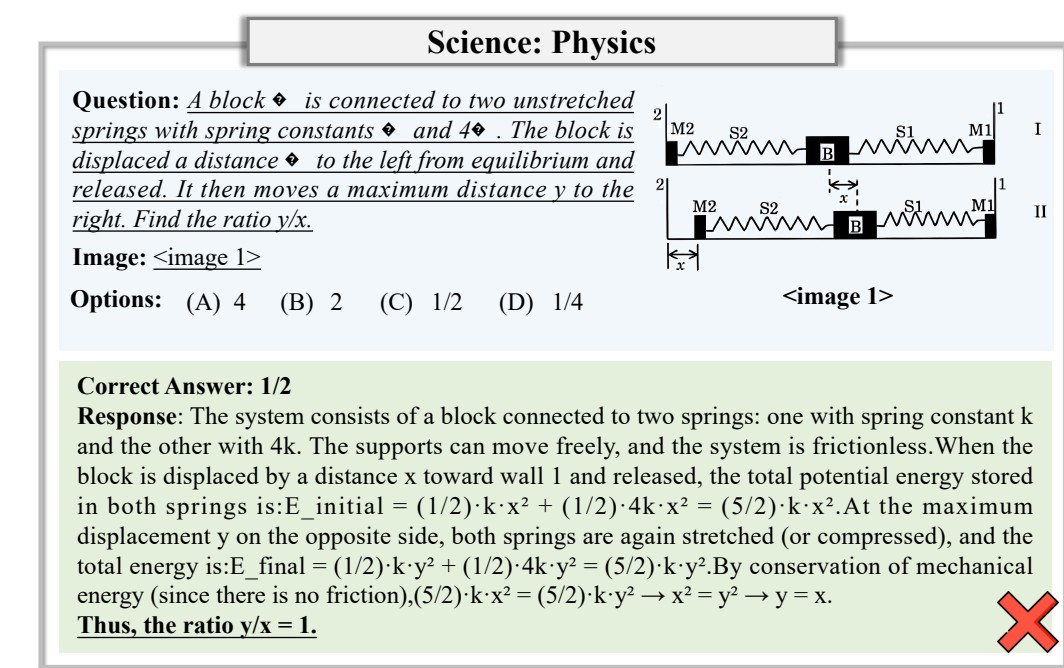

**Options:**   (A)  4    (B)  2    (C)  1/2    (D)  1/4

<image 1>

**Correct Answer: 1/2**

**Response**: The system consists of a block connected to two springs: one with spring constant k and the other with 4k. The supports can move freely, and the system is frictionless.When the block is displaced by a distance x toward wall 1 and released, the total potential energy stored in both springs is:E_initial = (1/2)·k·x² + (1/2)·4k·x² = (5/2)·k·x².At the maximum displacement y on the opposite side, both springs are again stretched (or compressed), and the total energy is:E_final = (1/2)·k·y² + (1/2)·4k·y² = (5/2)·k·y².By conservation of mechanical energy (since there is no friction),(5/2)·k·x² = (5/2)·k·y² → x² = y² → y = x.

**Thus, the ratio y/x = 1.**

Figure 26: Physics: Non-Thinking Case

**Science: Geography**

**Question:** *Which water in <image 1> coats soil particles? It's largely unavailable to plants, held tightly by strong molecular forces.*

**Image:** <image 1>

**Options:**

(A)  Gravitational water

(B)  Capillary water

(C)  Hygroscopic water

1 **Gravitational water**
When soil is saturated by rain, excess water is pulled through the soil by gravity and into groundwater.

2 **Capillary water**
When excess water is no longer flowing downward, the soil is at its field capacity. Water held around soil grains by capillary tension is available to plants.

3 **Hygroscopic water**
After a period without rain, when there is no more available water, the soil is at its wilting point. Water held by adhesion is not available to plants.

Field capacity

Wilting point

Decreasing soil moisture

<image 1>

**Correct Answer: (C) Hygroscopic Water**

**Response**: To identify the type of water that is tightly bound to soil particles and unavailable to plants, we examine how soil water is categorized. Water in soil exists in three main forms: gravitational, capillary, and hygroscopic. Hygroscopic water forms a very thin layer around soil particles and is held by strong adhesive forces. These forces are so intense that plants cannot extract this water, and it does not evaporate under normal conditions. In contrast, gravitational water drains quickly, and capillary water is available to plants.

**Therefore, the water that is firmly attached to soil particles and unavailable to plants is hygroscopic water.**

Figure 27: Geography: Non-Thinking Case

## Science: Biology

**Question:** *In pedigree <image 1> of a Gene A - related disease (non - shaded in - married individuals are non - carriers, complete penetrance), what is Individual 9's genotype?*

**Image:** <image 1>

**Options:**

(A) AA    (B) Aa    (C) Aa or AA    (D) $X^A X^A$

(E) $X^A X^a$    (F) $X^A X^A$ or $X^A X^a$    (G) $X^a X^a$

<image 1>

**Correct Answer: E**
**Response**: To determine the genotype of Individual 9, we analyze the pedigree and identify the mode of inheritance. The disease appears in every generation and affects both males and females, indicating an autosomal dominant pattern. In this pattern, affected individuals must carry at least one dominant disease allele (A), while unaffected individuals are homozygous recessive (aa). Individual 9 is not shaded, meaning they do not express the disease phenotype. This implies they do not carry the dominant allele and must have the genotype aa.
**Therefore, the correct genotype of Individual 9 is aa**.

Figure 28: Biology: Non-Thinking Case

