# OpenReview forum: "MMLU-Reason: Benchmarking Multi-Task Multi-modal Language Understanding and Reasoning"
_ICLR.cc/2026/Conference — ICLR 2026 Conference Withdrawn Submission_

### Official Review · Reviewer_mZBa · 2025-10-26

**Soundness:** 3
**Presentation:** 3
**Contribution:** 3
**Rating:** 6
**Confidence:** 4

**Summary:**

The paper shows MMMR, a new benchmark designed to evaluate the multi-modal reasoning capabilities of Large Language Models. The authors argue that existing benchmarks focus too much on the final answer's correctness and cannot analyze the quality of the reasoning process itself. To solve this, it consists of two main parts: (1) A high-difficulty dataset of 1,083 questions spanning six domains: Logic, Math, Code, Space-Time, Map-Plan, and Science. (2) A Reasoning Trace Evaluation Pipeline (RTEP), a novel framework to assess the quality of a model's "thinking" by measuring its relevance to the question (RTQ), relevance to the answer (RTA), and internal consistency (RSC). Key findings show that while "thinking" models like Gemini-2.5 Pro outperform non-thinking models, they still lag behind human-AI expert performance.

**Strengths:**

1. it shifts the evaluation from what the answer is to how the model arrived at it. This provides a much deeper understanding of model failures.
2. the RTEP framework (with metrics like RTQ, RTA, and RSC) offers a structured way to quantify reasoning quality beyond simple accuracy.
3. The benchmark's 1,083 questions are specifically designed to be challenging and require multi-step, multi-modal reasoning across six distinct and complex domains.
4. it categorizes the types of errors they make (e.g., "Inconsistency," "Overthinking," "Perceptual Error"), which gives researchers a clear roadmap for what to fix.

**Weaknesses:**

Experimental Limitations: The authors explicitly state (Section 2.3) that "Due to API restrictions, statistical significance tests were limited for closed-source models," which is a minor weakness in the completeness of the experimental analysis.

**Questions:**

It would be great if the authors can compare their methods to more existing benchmarks on MLLM reasoning and explain their unique contribution.

---

### Official Review · Reviewer_6Koq · 2025-10-31

**Soundness:** 2
**Presentation:** 2
**Contribution:** 2
**Rating:** 4
**Confidence:** 4

**Summary:**

This paper introduces MMMR, a new benchmark designed to evaluate the reasoning processes of Multi-Modal Large Language Models (MLLMs). It features 1,083 high-difficulty questions across six domains and a novel Reasoning Trace Evaluation Pipeline (RTEP). The pipeline assesses the quality of intermediate thinking steps using metrics like relevance and consistency, moving beyond simple answer accuracy. Empirical results show that even top-performing models exhibit flawed reasoning, such as inconsistency and overthinking. The work concludes that a significant gap exists between achieving correct answers and demonstrating sound reasoning fidelity.

**Strengths:**

1.  **Pioneering a Critical Evaluation Paradigm:** The paper's strength is its focus on evaluating the **reasoning process** rather than just the final answer correctness. It correctly identifies this as a major gap in existing MLLM benchmarks and proposes a structured way to address it. This shifts the conversation from "Did the model get it right?" to "How did the model arrive at its conclusion, and was the process sound?"

2.  **Conceptual Innovation of the Reasoning Trace Evaluation Pipeline (RTEP):** The introduction of the RTEP and its core metrics—Relevance to Question (RTQ), Relevance to Answer (RTA), and Reasoning Step Consistency (RSC)—is a strong conceptual contribution. It provides a new vocabulary and a structured framework for analyzing the quality of intermediate thought processes, which can be adopted and refined by future research.

3.  **Well-Defined Error Taxonomy:** The paper introduces a granular and insightful taxonomy for both "Thinking Errors" (e.g., Inconsistency, Overthinking) and "Answer Errors" (e.g., Reasoning Error, Perceptual Error). This classification provides a much-needed diagnostic tool that allows researchers to move beyond simple accuracy scores and pinpoint specific failure modes in models.

**Weaknesses:**

1.  **Misleading Title:** The benchmark contains 1,083 questions in total (977 in the test set). In the context of modern LLM benchmarks (e.g., MMLU with >14k questions, Big-Bench with >200 tasks), this size is small. A smaller, high-quality, deeply annotated dataset is valuable but still is small.

2.  **Limited Scope of Analysis:** The most novel and interesting analyses are presented as case studies on a very small number of models.
    *   **Thinking Quality Analysis (Table 4):** The detailed comparison of reasoning traces is only performed between `Claude-3.7-sonnet` and the custom `Dual` model. While the finding is interesting (higher accuracy can correlate with worse reasoning quality), making a general claim based on just two models is a significant overreach. This analysis should have been applied to all MLLMs-T to be convincing.
    *   **Error Analysis (Figure 5):** The distribution of thinking and answer errors is only shown for `Claude-3.7-sonnet`. Different models and architectures likely have different failure modes. For instance, a model with weaker vision capabilities might have more *Perceptual Errors*, while another might have more *Inconsistency*. Without a comparative error analysis, the insights remain specific to one model rather than the field.

3.  **Potential for Overfitting and High Variance:** With only 977 test questions spread across six distinct and complex domains (an average of ~163 questions per domain), the benchmark is susceptible to high variance. A model's performance could be heavily influenced by its fit to the specific style of questions in this small sample, rather than its general reasoning ability. This small size also makes it easier for future models to "overfit" to the benchmark through targeted tuning.


4.  **Lack of Transparency in Dataset Curation:** This is a major methodological flaw.
    *   The paper states the dataset is "rigorously curated" but does not provide details on the actual process. It cites several other papers as sources (lines 071-075) and mentions "Web, Textbook, Remake" (Figure 3), but the process remains opaque.
    *   **"Remake" Process:** 44.6% of items are described as "remade or enhanced" (line 247). How was this done? Was it to increase difficulty, remove artifacts, or rephrase? Without a clear methodology, it's impossible to assess the quality or potential biases introduced during this process.
    *   **Data Contamination:** The use of web sources raises concerns about data contamination. Have the authors checked if questions or similar examples appear in the training sets of the models being evaluated? This is a missing step.



5.  **Over-reliance on LLM-as-a-Judge:** The core contribution, the Reasoning Trace Evaluation Pipeline (RTEP), relies on GPT-4o as an automated evaluator for metrics like RTQ, RTA, and RSC. This approach has several known issues:
    *   **Inherent Biases:** LLM judges are known to have biases, such as preferring longer, more verbose answers (which seems to be penalized by their `TLen` metric, creating a potential conflict), favoring certain stylistic patterns, or position bias.
    *   **Insufficient Validation:** The authors state they validated the judge against human annotators on only **50 traces** (line 363). For a dataset of 1,083 questions and 17 models, 50 samples is a fraction and likely insufficient to prove the judge's reliability across all six diverse domains and various model output styles. The reported 88% agreement is good, but its generalizability is questionable.
    *   **Reproducibility:** The exact prompts and configuration used for the GPT-4o judge are crucial for reproducibility but are not fully detailed. The quality of an LLM judge is highly sensitive to prompting.



6.  **Arbitrary Metric Formulation:** The "Overall Score" (OS) in Table 4 is defined by a weighted sum: `0.3*RTQ + 0.3*RTA + 0.3*RSC + 0.1*(ACC*0.1)`. The choice of these weights (0.3 for each trace metric, but a tiny 0.01 for accuracy) is not justified. This weighting heavily favors reasoning "quality" over correctness, which may be the paper's goal, but the specific values could be tuned to support a desired outcome, making the metric less objective.

**Questions:**

1.  **Curation and "Remake" Process:** What was the specific, step-by-step protocol for curating the dataset? For the 44.6% of items that were "remade or enhanced," what exact changes were made and what were the guiding principles for these modifications?

2.  **Data Contamination:** What measures were taken to ensure that the questions, particularly those sourced from the web, were not present in the training data of the evaluated models? Was any decontamination process performed?


3.  **Reproducibility of the LLM Judge:** Can the exact prompts, API parameters (like temperature), and rule-based checklists used to guide the GPT-4o judge for the RTQ, RTA, and RSC metrics be provided to ensure full reproducibility?


4. **Definition of Error Categories:** How were the thinking and answer error categories (e.g., "Inconsistency," "Overthinking") formally defined? Was there a detailed annotation guide, and what was the inter-annotator agreement among human experts for this classification task?

---

### Official Review · Reviewer_WbKc · 2025-11-01

**Soundness:** 3
**Presentation:** 2
**Contribution:** 4
**Rating:** 4
**Confidence:** 3

**Summary:**

The MMMR benchmark is introduced to rigorously assess the multi-modal reasoning capabilities of large language models, particularly those with intermediate thinking processes (MLLMs-T). Unlike traditional benchmarks, which focus mainly on answer accuracy, MMMR evaluates how models reason across six domains: Logic, Math, Code, Map, Science, and Space-Time. The dataset includes 1,083 high-difficulty tasks with complex reasoning demands and multi-modal input, such as images and text. A unique feature of MMMR is its Reasoning Trace Evaluation Pipeline (RTEP), which not only checks answer correctness but also assesses the quality and logical consistency of the model's reasoning steps. Empirical results reveal that, while MLLMs-T models outperform standard models in reasoning tasks, they still struggle with issues like logical inconsistency and overthinking, highlighting gaps between performance and reasoning quality.

**Strengths:**

1.	The introduction of MMMR as a new benchmark for multi-modal reasoning is a significant contribution. By focusing on reasoning depth rather than just accuracy, the authors address an important gap in the evaluation of Multi-Modal Large Language Models (MLLMs).
2.	The Reasoning Trace Evaluation Pipeline (RTEP) is a novel addition, offering insights into the reasoning process, providing valuable metrics for understanding the coherence, consistency, and relevance of thinking traces, which is a key advancement.
3.	The multi-modal dataset spans diverse domains (Logic, Math, Space-Time, Code, Map, Science), which ensures that the benchmark is comprehensive and applicable across various problem types.
4.	Empirical results and comparisons with state-of-the-art models are well presented, showing the effectiveness of MLLMs-T in comparison to non-thinking counterparts.

**Weaknesses:**

1. There is a lack of clear discussion on how to address or mitigate the reasoning pathologies such as overthinking and inconsistency, which are frequently observed in the models evaluated.
2.  Issues with Figure Layout:
The layout of the figures in the paper is problematic. The arrangement of the diagrams/figures does not appear to be optimal and affects the overall presentation. It would be beneficial to revise the figure placements and spacing to ensure clarity and improve the visual flow of the paper.
3. More detail on how the RTEP pipeline can be automated or scaled up for other benchmarks could enhance its broader applicability in future research.

**Questions:**

Please see question.

---

### Official Review · Reviewer_X2NN · 2025-11-03

**Soundness:** 3
**Presentation:** 2
**Contribution:** 3
**Rating:** 2
**Confidence:** 4

**Summary:**

The paper introduces MMMR, a new benchmark for evaluating the reasoning capabilities of Multi-Modal Large Language Models (MLLMs) and their “thinking” variants (MLLMs-T). It contains 1,083 high-difficulty tasks across six domains—logic, math, space-time, code, map, and science—designed to test multi-hop and symbolic reasoning. The authors propose a Reasoning Trace Evaluation Pipeline (RTEP) that measures reasoning quality using metrics such as relevance to the question (RTQ), relevance to the answer (RTA), and reasoning step consistency (RSC). Experimental results show that MLLMs-T outperform standard MLLMs, but even top models like Gemini-2.5 Pro still trail human-level reasoning by over 10%. Analysis reveals that reasoning errors such as inconsistency, overthinking, and irrelevant traces remain pervasive. Overall, MMMR provides a scalable and interpretable framework for diagnosing and improving the reasoning processes of next-generation multi-modal models.

**Strengths:**

1. Comprehensive and High-Difficulty Benchmark Design: MMMR covers six diverse reasoning domains with carefully curated, high-complexity tasks, ensuring broad and deep evaluation of multi-modal reasoning. Its inclusion of symbolic, spatial, and scientific reasoning makes it a strong diagnostic tool beyond perception-based benchmarks.

2. Innovative Evaluation Pipeline for Reasoning Quality: The proposed Reasoning Trace Evaluation Pipeline (RTEP) introduces structured metrics—RTQ, RTA, and RSC—to assess reasoning coherence rather than only answer correctness. This design enables systematic analysis of how models think, not just what answers they produce.

3. Clear Empirical Insights and Diagnostic Value: The experiments compare 17 models and reveal key weaknesses like inconsistency and overthinking, offering concrete directions for improving multi-modal reasoning. The benchmark thus provides both quantitative and qualitative insights that can guide future model development and evaluation.

**Weaknesses:**

1. As a key evaluation component of this paper, the description of how RTA, RTQ, and RSC scores are determined lacks sufficient detail and example prompts. The statement “each rated on a 0–10 scale. Scores combine rule-based checklists and semantic checks. Leveraging GPT-4o as an automated evaluator” is overly brief and does not clearly specify the evaluation criteria.
2. The paper does not provide enough information on the data collection process, data sources, or filtering standards, raising concerns about the overall quality and reliability of the benchmark questions.
3. There are noticeable formatting issues in the paper layout, suggesting a rushed preparation—for example, Table 2 is incomplete, and there are blank on page 5.
4. Some descriptions of the dataset’s characteristics appear overly subjective, such as “many questions require long-horizon reasoning, abstraction, or visual-spatial synthesis,” without providing the corresponding data sources, annotation standards, or quantitative examples from model reasoning to substantiate the claimed difficulty.
5. The experimental results concerning the paper’s core contribution, RTEP, are insufficient and lack comparisons with Chain-of-Thought (CoT) reasoning models, limiting the generalizability of the findings. Moreover, the rationale for selecting Claude-3.7-Sonnet and GPT-4V+DeepSeek-R1 as comparative baselines is unclear.
6. The paper lacks qualitative case analyses of model output traces, making it difficult for readers to intuitively understand what kinds of reasoning behaviors satisfy or violate the proposed fine-grained evaluation metrics.

**Questions:**

1. Some MLLM candidates (e.g., Qwen2.5-VL) are capable of performing reasoning under Chain-of-Thought (CoT) prompting, so presenting only their direct answer results is incomplete.
2. The paper lacks evaluation on open-source vision-language models (VLMs) that have benefited from reinforcement learning techniques; including these as references for community “thinking” methods would strengthen the conclusions.
3. The paper claims to present “the first evaluation pipeline for thinking of MLLMs-T,” yet prior work (e.g., *MME-CoT: Benchmarking Chain-of-Thought in Large Multimodal Models for Reasoning Quality, Robustness, and Efficiency*) has already proposed fine-grained evaluations of intermediate reasoning processes. Since some of these methods share overlapping evaluation metrics, the authors should provide a clearer explanation of how MMMR differs from existing approaches and what unique contributions it offers.

---

### Note · Authors · 2025-11-14

I have read and agree with the venue's withdrawal policy on behalf of myself and my co-authors.